# RNA aptamers specific for transmembrane p24 trafficking protein 6 and Clusterin for the targeted delivery of imaging reagents and RNA therapeutics to human β cells

Dimitri Van Simaeys [1,5], Adriana De La Fuente [1,5], Serena Zilio [1,5], Alessia Zoso [2], Victoria Kuznetsova[1], Oscar Alcazar[2], Peter Buchwald [2], Andrea Grilli [3], Jimmy Caroli [3], Silvio Bicciato[3] & Paolo Serafini[1,2,4✉]

The ability to detect and target β cells in vivo can substantially refine how diabetes is studied and treated. However, the lack of specific probes still hampers a precise characterization of human β cell mass and the delivery of therapeutics in clinical settings. Here, we report the identification of two RNA aptamers that specifically and selectively recognize mouse and human β cells. The putative targets of the two aptamers are transmembrane p24 trafficking protein 6 (TMED6) and clusterin (CLUS). When given systemically in immune deficient mice, these aptamers recognize the human islet graft producing a fluorescent signal proportional to the number of human islets transplanted. These aptamers cross-react with endogenous mouse β cells and allow monitoring the rejection of mouse islet allografts. Finally, once conjugated to saRNA specific for X-linked inhibitor of apoptosis (XIAP), they can efficiently transfect non-dissociated human islets, prevent early graft loss, and improve the efficacy of human islet transplantation in immunodeficient in mice.

[1] Department of Microbiology and Immunology, Miller School of Medicine, University of Miami, Miami, FL, USA. [2] Diabetes Research Institute, Miller School of Medicine, University of Miami, Miami, FL, USA. [3] Center for Genome Research, Department of Life Sciences, University of Modena and Reggio Emilia, Modena, Italy. [4] Sylvester Comprehensive Cancer Center, Miller School of Medicine, University of Miami, Miami, FL, USA. [5] These authors contributed equally: Dimitri Van Simaeys, Adriana De La Fuente, Serena Zilio. ✉email: pserafini@miami.edu

The identification of specific probes against human β cells is an unmet medical need that would allow accurate, non-invasive visualization of β-cell mass in vivo, early identification of promising experimental therapeutic approaches, development of tools for the early diagnosis of Type 1 diabetes (T1D), and targeted delivery of therapeutics to β cells in vivo[1–3]. Recent preclinical studies suggest that impaired β cells function is an early feature of T1D and T2D pathogenesis while a substantial decrease in β-cell mass, characterized by phases of relapsing and remitting progression, occurs more closely to clinical manifestation[4,5]. However, several technical limitations hamper a thorough investigation of human β cell mass during the asymptomatic time that precedes the clinical manifestation of diabetes[6,7]. Although positron emission tomography (PET), single-photon emission computed tomography (SPECT), and magnetic resonance imaging might be used to measure β-cell mass in humans, the absence of optimal β cells specific probes restricts the clinical use of these techniques to islets transplantation settings[8–10]. Clinical attempts to target β cells have been made using probes against molecules and pathways found over-expressed by transcriptome analysis on β cells. For example, probes targeting the type 2 vesicular monoamine transporter (VMAT2)[11–13] or the glucagon-like peptide-1 receptor (GLP-1R)[14,15] demonstrated reduced PET/SPECT signals in T1D patients compared to the healthy control. However, these approaches may not be sufficiently sensitive to assess the relatively small changes in β cell mass that may occur longitudinally in patients with diabetes[7], and β-cell GLP1R is itself down-regulated in diabetes[16] making it a less attractive target for drug delivery. Additionally, the number of target molecules expressed on the membrane of β cells represents a critical limit for the delivery and cellular uptake of imaging reagents and therapeutics.

The ideal β-cell probe(s) for imaging and therapeutic delivery should be nontoxic, non-immunogenic, and able to penetrate deep into the tissues to bind specific target(s) highly expressed on β cells[17]. We hypothesized that the combinatorial use of RNA aptamers, each targeting a different epitope on β cells, might be used to measure β-cell mass and for the targeted delivery and intracellular accumulation of therapeutics.

RNA aptamers are small non-immunogenic oligonucleotides that bind to their ligand with high affinity[18–20]. They can be generated against known and unknown molecules expressed on the desired cell type by a selection process called Systematic Evolution of Ligands by EXponential enrichment (SELEX)[21]. Once identified, aptamers can be chemically synthesized and easily engineered during synthesis to modify their bioavailability, pharmacokinetics, and/or function[22]. These properties support rapid, scalable, low-cost reproducible aptamer production that simplifies manufacturing approval for human use[23].

Here, we report the identification and characterization of two RNA aptamers, with putative specificity for TMED6 and clusterin, that recognize both human and mouse β cells with high selectivity in vitro and in vivo. These aptamers allow the quantitative measurement of human β-cell mass after engraftment in immune deficient mice and during the rejection of islet allografts in immunocompetent mice. These aptamers also allow the efficient and non-viral delivery of saRNA to upregulate the antiapoptotic gene *XIAP* in human non-dissociated islets. This significantly increases the efficiency of islets transplantation by inhibiting early graft loss.

## Results

**Selection of aptamer libraries against human islets**. We used two independent and unsupervised selection approaches to identify aptamers specific for human islets: high-throughput (HT-) cluster-Cell-SELEX and HT-Toggle SELEX[24]. The first approach can isolate aptamers against all membrane proteins expressed by human islets. Still, its efficacy can be compromised by the viability of the preparation and the ischemic time of the organ. The toggle cell SELEX is designed to isolate aptamers starting from freshly isolated, highly viable mouse islets and allows for selecting aptamers that cross-react between mouse and human islets.

In the HT-cell SELEX approach (Fig. 1a), a library of ~$10^{14}$ random RNA sequences underwent iterative negative and positive selection cycles using islet-depleted acinar tissues and hand-picked, non-dissociated human islets, respectively. Eight selection cycles were performed using islets and acinar preparations from four different cadaveric donors. The selection stringency was gradually increased after cycle four by increasing the number of washes and decreasing the number of islets used in the selection. Libraries from each cycle were sampled, sequenced, and analyzed using Clustal-Ω and APTANI[25,26]. After each cycle, we observed a decrease in library complexity (Fig. 1b), indicating a successful enrichment process. Clustal-Ω showed the presence of 208 clusters of similar sequences in the library after cycle 8 (Fig. 1c). Thirty-one of these clusters had a cumulative frequency higher than $10^{-4}$ and accounted for 66.7% of the sampled library (Fig. 1c and Supplementary Table 1). Despite the substantial enrichment by HT-cell SELEX, the selected polyclonal aptamer library failed to recognize human islets in fluorescence microscopy (not shown), possibly because the concentration of islet-specific aptamers was too small to provide sufficient fluorescent signal.

In the toggle cell-SELEX strategy (Fig. 1d), eight selection cycles were initially performed using islet-depleted mouse acinar tissue and handpicked mouse islets as negative and positive selectors, respectively. The resulting library recognized islets in mouse pancreas sections but not human counterparts (Fig. 1e). We then toggled the selection from mouse to human and performed two additional SELEX cycles using human islets and acinar tissue as positive and negative selectors, respectively (Fig. 1d). Strikingly, these two selection cycles performed with human tissues were sufficient to generate a polyclonal library of aptamers capable of identifying islets in sections of human pancreases. In contrast, no staining was observed with a random control library (Fig. 1e, f). This suggested an important enrichment of those aptamers specific to human islets.

The cycle 8 library from the selection using mouse islets (M8) and the library obtained from the two additional selections with human islets (M8H2) were sequenced, and aptamers frequencies were compared (Fig. 1g). We observed that 1172 aptamers with a frequency higher than the background ($10^{-6}$) expanded more than 16 times during the two selection rounds with human specimens. Additionally, we detected 106 aptamers (frequency > $10^6$) in M8H2 but not in M8. We then assumed that islet-specific aptamers must be present in sufficient amounts to give a detectable signal in fluorescence microscopy (i.e., Fig. 1f) and that their frequency should have increased during the two selection rounds with human islets. Therefore, we focused only on those aptamers whose frequency in M8H2 was at least $10^{-4}$ and were either undetected in M8 or expanded at least ten times with the last two rounds of selection with human islets. Additionally, we chose aptamers largely represented in M8 (frequency > $10^{-4}$) that at least doubled in frequency after the selections with human tissues. This process resulted in the choice of 39 aptamers (Supplementary Table 2).

We then evaluated whether the two selection strategies resulted in the isolation of similar aptamers. In particular, the most frequent sequences from the 31 most frequent families from the cluster-cell SELEX were analyzed in Clustal-Ω together with 39 sequences chosen from the toggle SELEX. This analysis revealed that identical or very similar aptamers were isolated independently using the two approaches (Fig. 1h). Taken

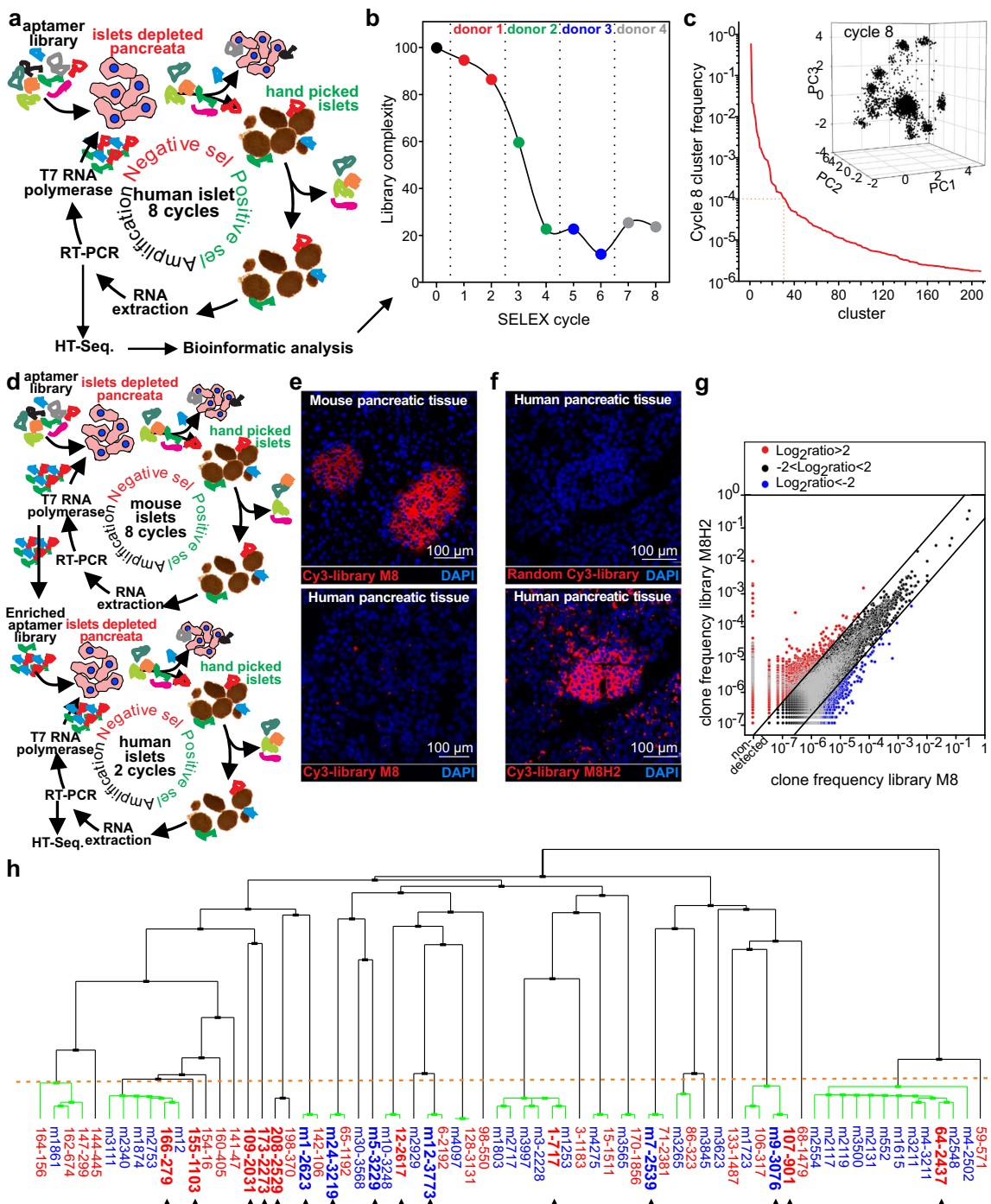

**Fig. 1 Cluster-cell SELEX and Toggle SELEX allowed the convergent selection of polyclonal aptamers against human islets. a** Schematic diagram of HT Cluster-cell SELEX. A random library of ~$10^{14}$ random aptamers was negative and positive selected against islet-depleted exocrine cells and islets from cadaveric donors. Aptamers were then recovered, amplified, and used for the following selection cycle. Pancreatic specimens from four cadaveric donors were used for the eight selection cycles. **b** Library complexity decreased during SELEX. Sampled cDNA from the different cycles underwent Illumina-based HT-sequencing, and data were analyzed for aptamer frequency, library complexity, and families identification. **c** Frequency of the clusters identified by Clustal Ω from cycle 8 library and the sum of the frequencies of aptamers from each of the 208 identified families were plotted. The insert shows the principal component analysis of the 5000 most frequent sequences. **d** Schematic diagram of HT-Toggle SELEX. Eight cycles (M1–M8) of selection were performed using acinar cells and islets isolated from BALB/c mice. Two additional cycles of selection (M8H1 and M8H2) were then performed using acinar cells and islets from one cadaveric donor. cDNA library from cycle M8 and M8H2 underwent HT-sequencing. **e** Mouse and human pancreatic specimens were stained with 1 μg of cy3-labeled libraries from cycle M8. **f** Human pancreatic tissues were stained with cy3-library from cycle 0 (irrelevant control) or M8H2. **g** Frequency of aptamers from cycle M8 and cycle M8H2. **h** Neighbor-joining tree of selected aptamers from the two selection strategies. Thirty-one aptamers, chosen from the "cluster-cell" SELEX (red), and 39 aptamers, from the toggle SELEX (blue), were aligned. Branches that contain aptamers from both selections are highlighted in green. Fifteen aptamers (arrows) representing the main branches were selected for empirical testing. Source data are provided as a Source Data file.

together, these data suggest that both unsupervised SELEX approaches successfully generated libraries enriched in aptamers specific for human islets.

**Aptamer 1-717 and m12-3773 are specific for human islets and preferentially bind β cells.** Based on the Clustal-Ω analysis comparing the libraries from the two SELEX approaches, the 15 monoclonal aptamers (Supplementary Fig. 1) highlighted in Fig. 1h were selected for empirical validation as representative of the main tree branches. Aptamers were synthesized from corresponding oligonucleotides (Supplementary Table 3) by PCR and T7-RNA polymerase, cy3-labeled, purified, and used as probes in immune fluorescence microscopy against human pancreatic tissues (Fig. 2a). Cycle 0 and M8H2 libraries were used as negative and positive controls, respectively. Twelve of the 15 tested aptamers recognized islets in the tissue; however, only seven (highlighted in yellow in Fig. 2a) of these did not recognize or showed low binding to the surrounding acinar tissue. Since islet-specific binding in the pancreas does not exclude binding to other tissues, we tested these aptamers on FDA-approved tissue arrays (Fig. 2b, Supplementary Fig. 2). Each of these arrays contains 30 tissues with replicas from three healthy donors allowing the simultaneous staining and comparison of aptamers binding to different human tissues. Tissue arrays were stained with cy3-labeled aptamers, counterstained with DAPI and anti-insulin antibodies, and images were acquired and processed with Cell Profiler[27]. As expected, all tested aptamers showed binding to pancreatic islets, but five of them also recognized other tissues with high intensity. Importantly, aptamers m12-3773 and 1-717 recognized the islets with significantly higher intensity ($P_{anova} < 0.001$) than all other tissues (MFI$_{islets}$ of 6.55 ± 1.24 and 10.42 ± 3.53 vs. MFI$_{other\ tissues}$ of 1.25 ± 1.31 and 1.3 ± 1.86 for aptamers m12-3773 and 1-717, respectively, Fig. 2b and Supplementary Fig. 2).

The specificity of these two aptamers within the islets was then evaluated by fluorescence microscopy and image cytometry on human pancreatic sections (Fig. 2c, d, and Supplementary Figs. 3–5), by confocal microscopy and image cytometry (Fig. 2e Supplementary Fig. 3c), and by flow cytometry on dissociated human islets (Fig. 2f). Image cytometry analysis of whole scanned sections of human pancreases further confirmed the specificity of the two aptamers for the islets over the acinar tissues. However, aptamer m12-3773 showed some binding to cells in the close proximity of the islets. Only background signal was observed with scrambled aptamers (Fig. 2d and Supplementary Fig. 3). Of note, the use of biotinylated aptamers and optimized blocking buffer reduced the non-specific and highlighted the specificity of both aptamers for the islets (Fig. 2c and Supplementary Fig. 4).

These approaches revealed preferential binding of aptamers m12-3773 and 1-717 to insulin-producing β cells. Aptamer binding showed similar specificity across race, gender, and BMI (see Supplementary Table 4 for donor demographic) with apparent $K_D$ for MIN6 insulinoma in the nanomolar range (Supplementary Fig. 6). Taken together, these data indicate that aptamers m12-3773 and 1-717 are highly specific for human islets and preferentially bind to β cells. However, we did observe some binding to a few alpha cells and a few acinar cells neighboring islets and in the pancreatic ducts.

**Clusterin and TMED6 are the cognate targets of aptamer m12-3773 and 1-717, respectively.** Aptamer-based immune precipitation followed by mass spectrometry was first employed to identify the cognate targets of aptamers m12-3773 and 1-717 (Fig. 3a). Briefly, we incubated single-cell suspension from human islets with islet-specific or scrambled biotinylated aptamers, extensively washed the cells, gently solubilized the cell membrane,

recovered the aptamers bound to their target with streptavidin magnetic beads, ran an SDS page, and evaluated the differentially expressed proteins by Coomassie blue staining. We observed a band of approximately 75kd in the immune precipitate from aptamer m12-3773 that was absent in precipitate from the scrambled aptamer. No differentially present bands were found in the precipitate from aptamer 1-717. This band and the corresponding region in the scrambled lane were cut and analyzed by mass spectrometry to detect differentially expressed proteins. Mascot analysis showed various nuclear and/or positively charged proteins in the immune precipitates from both the scrambled and m12-3773 aptamers (Supplementary Table 5). We detected seven peptides covering 25% and 39% of clusterin isoform 1 and isoform 3, respectively (Fig. 3a and Supplementary Table 5) in the immune precipitate from aptamer m12-3773. In contrast, we did not find any peptide from this protein in the immune precipitate from the scrambled aptamer. To test whether clusterin was the ligand of aptamer m12-3773, we took advantage of aptamer internalization properties and the available algorithm for using RNA aptamers as transfection reagents[28]. We used aptamer 1-717 conjugated with clusterin-specific siRNA or scrambled siRNA to transfect non-dissociated human islets. Forty-eight hours later, we employed one part of the islets to evaluate clusterin expression by qRT-PCR and another part to assess the binding of aptamer m12-3773. Treatment with siRNA$_{CLUS}$/apt#1-717 chimera, but not with siRNA$_{scrambled}$/apt#1-717 aptamer chimera significantly reduced clusterin expression on the islets (Fig. 3b). This downregulation correlated to significant inhibition of aptamer m12-3773 binding to the β cells as assessed by flow cytometry (Fig. 3c, d). Aptamer-chimera m12-3773 with Cy5-labeled guide RNA showed an affinity in the nanomolar range against MIN6 cells (Fig. 3e). This affinity was further increased using tetrameric aptamer m12-3737 (Supplementary Fig. 6b).

Immune precipitation and mass spectrometry did not identify the putative target for aptamer 1-717. While mass spectrometry on aptamer-based immunoprecipitation can identify targets in their native form, technical difficulties associated with the immunoprecipitation procedures and limits of current mass spectrometry analysis software often restrain the identification of putative binders[29]. We, therefore, developed an alternative strategy based on aptamer hybridization to high-density recombinant protein arrays. Protein microarrays simultaneously screen the binding of a ligand against 23,040 recombinant proteins spotted per array, avoiding the issues related to the solubilization of membrane proteins. We hybridized protein arrays with the aptamers and evaluated the differential binding of aptamer 1-717 and m12-3773 to the spotted proteins. We focused our analysis on the 3881 proteins whose genes are significantly upregulated (fold change >2, FDR < 0.01) in the islets compared to the whole pancreas (Supplementary Data File 1). TMED6 (p24γ5, UniProt: Q8WW62) appeared to be the putative ligand of aptamer 1-717 (Fig. 4a).

To test whether TMED6 was the target of aptamer 1-717, we performed a cold target inhibition assay in which cy3-labeled apt#1-717 was admixed at different ratios with unlabeled recombinant (r) TMED6 before staining of human pancreatic sections (Fig. 4b, Supplementary Fig. 7). The addition of rTMED6, but not a 20-fold excess of BSA, significantly inhibited aptamer 1-717 binding in a dose-dependent manner (Fig. 4b, Supplementary Fig. 7). Surface plasmon resonance analysis revealed a strong affinity ($k_a = 7.1 \times 10^4 \pm 5 \times 10^1\ M^{-1}\ s^{-1}$, $k_d = 4.6 \times 10^{-4} \pm 1.4 \times 10^{-6}\ s^{-1}$, $K_D = 6.5 \times 10^{-9} \pm 2.5 \times 10^{-11}\ M$) of apt#1-717 for TMED6 (Fig. 4c). Of note, this affinity is much higher than the one observed for primary β cells (Supplementary Fig. 6a), possibly because of the presence of dead cells and because the cy3 fluorochrome randomly conjugated in the aptamer may interfere with the binding. Indeed, the use of

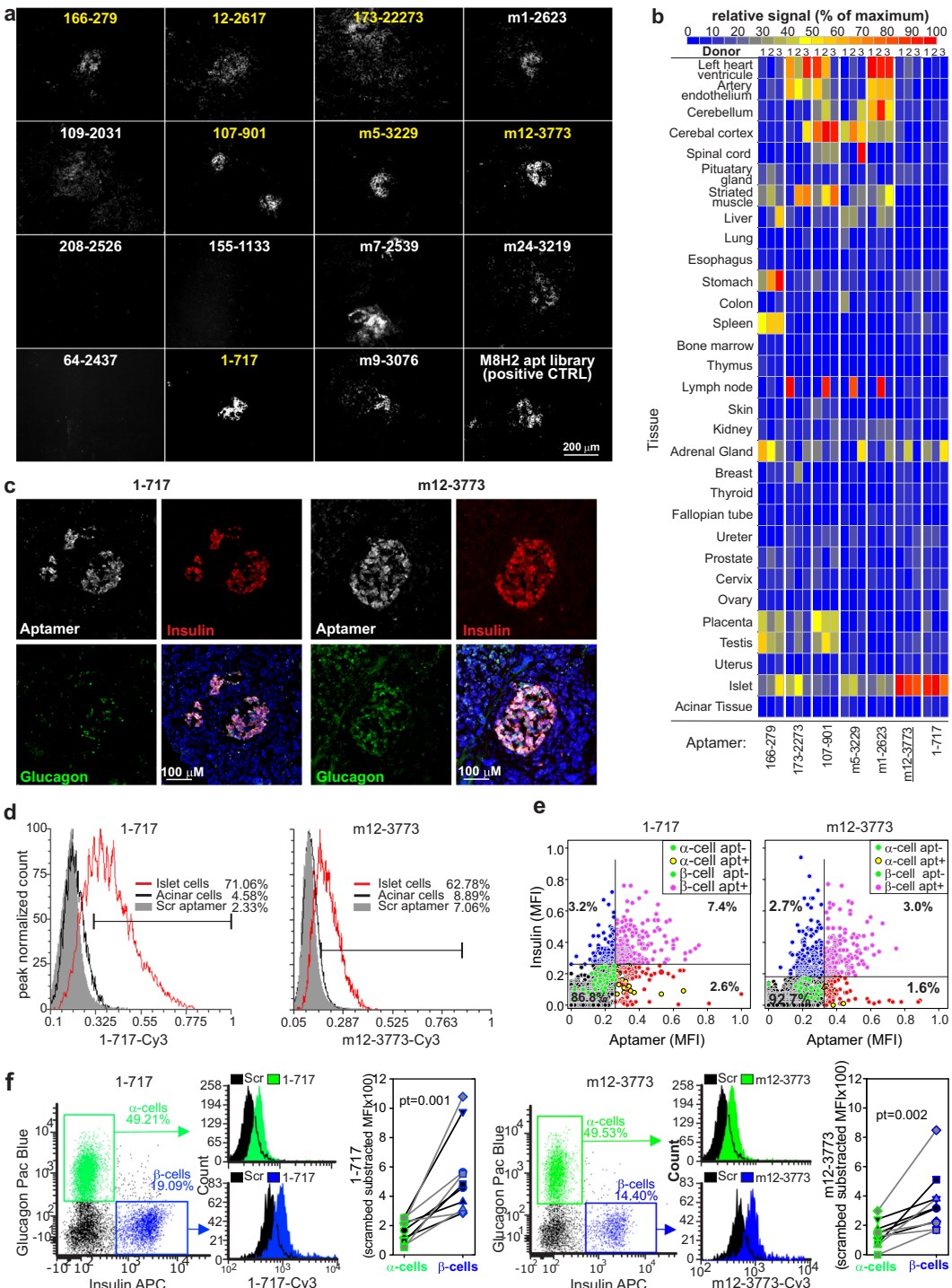

**Fig. 2 Aptamer 1-717 and m12-3773 show high specificity for human islets. a** Fifteen aptamers from the two SELEX strategies were produced, Cy3 labeled and used as immunofluorescence probes on human pancreatic tissues. One experiment representative of two is depicted. **b** The seven aptamers with the highest specificity for the islets (highlighted in yellow in panel a) were tested in FDA-approved tissue arrays against 30 different tissues from three different donors. Samples were counterstained with anti-insulin antibodies and images analyzed by Cell Profiler. Integrated fluorescence intensity resulting from all the cells or insulin-positive cells were fed into http://jcolorgrid.sourceforge.net/, and data are expressed as a percentage of highest intensity. Representative images from the different tissues are shown in Supplementary Fig. 2. **c** Sections of human pancreatic tissue stained with aptamer chimera 1-717 and m12-3773 hybridized with cy5 guide RNA and with antibodies against insulin and glucagon were acquired by fluorescence microscopy. One experiment representative of the other five is shown. **d** Scanned images of all pancreatic sections stained against insulin and glucagon and with aptamer 1-717, aptamer m12-3773, or scrambled aptamers were analyzed with Cell Profiler and FCS express. Median fluorescence intensity (MFI) from islets and acinar tissue gates (Supplementary Fig. 4) is shown. **e** Images from 4 to 5 islets and surrounding tissues were acquired by confocal microscopy, processed by Cell Profiler and MFI per single cells plotted. Spearman correlation: $r = 0.639$, $p = 2 \times 10^{-7}$, and $r = 0.225$, $p = 2 \times 10^{-7}$ for aptamer 1-717 and m12-3773, respectively. **f** Single-cell suspension from islet preparation from cadaveric donors ($n = 9$) were stained with cy3-labeled aptamer 1-717 or m12-3773, live dead dye, anti-insulin, and anti-glucagon antibodies, and analyzed by flow cytometry. Paired $T$ test value is shown. Source data are provided as a Source Data file.

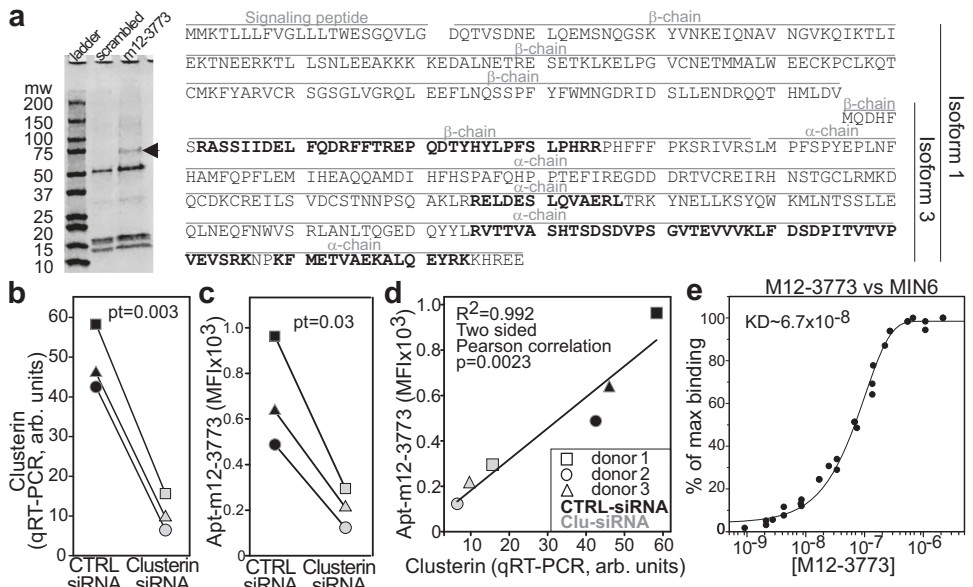

**Fig. 3 Clusterin is the putative target for aptamer m12-3773. a** Single-cell suspension from human islets was incubated with biotinylated aptamer m12-3773 or scrambled aptamer, cell membrane lysed, and aptamer ligand complexes were isolated with magnetic beads conjugated with streptavidin. Captured proteins underwent SDS page and Coomassie blue staining. The differentially expressed band from m12-3773 (arrow) or the corresponding gel area from the scrambled immune precipitate was extracted and analyzed by mass spectrometry. Peptides identified by the Mascott software are in bold in the clusterin sequence. One experiment representative of other two is shown. **b** Non-dissociated human islets were transfected with aptamer-chimera composed of aptamer 1-717 and either siRNA specific for clusterin or scrambled siRNA. Clusterin expression was evaluated 72 h later by qRT-PCR. Two sides paired. T test was performed. **c** Islets from (**b**) were dissociated, stained with cy3-aptamer m12-3773, and antibodies against insulin and glucagon. MFI after gating on viable insulin-positive cells is shown. Two sides paired. T test was performed. **d** Expression of clusterin evaluated via qRT-PCR (**b**) was plotted against the MFI of m12-3773 on β cells (**c**). **e** Affinity. MIN6 cells were stained with vital dye and different concentrations m12-3773 aptamer-chimera annealed with cy5-labeled guide saRNA and analyzed by flow cytometry. Source data are provided as a Source Data file.

aptamer-chimera hybridized with cy5-conjugated guide RNA showed affinity in the nanomolar range (Fig. 4d). As expected, the use of aptamer conjugated to streptavidin (i.e., tetrameric aptamers) further increases the affinity/avidity of 1-717 to MIN6 cells (Supplementary Fig. 6b) to a low nanomolar range.

To further evaluate if TMED6 was the target of aptamer 1-717, we transfected non-dissociated human islets with an aptamer-chimera composed of aptamer m12-3773 and TMED6-specific siRNA (siRNA$_{TMED6}$). Compared to the scrambled siRNA, aptamer-chimera mediated TMED6 silencing significantly inhibited both TMED6 expression evaluated by qRT-PCR (Fig. 4e) and the binding of aptamer 1-717 to the islets assessed by flow cytometry (Fig. 4f, g). Taken together, these experiments strongly suggest that clusterin and TMED6 are the cognate targets of aptamer m12-3773 and 1-717, respectively.

**Aptamer 1-717 and m12-3773 allow in vivo quantification of human β cells**. After establishing the specificity and identifying the cognate targets of aptamers 1-717 and m12-3773, we evaluated whether these aptamers could recognize human islets in vivo. We engrafted human islets under the kidney capsule of immunodeficient NOG mice and, 21 days later, injected the mice with either one (12.5 pmoles/g) or both aptamers (6.25 pmoles/g each) intravenously. We assessed aptamer biodistribution 24 h later by qRT-PCR (Fig. 5a). Both aptamers 1-717 and m12-3773 showed preferential accumulation in the islet grafts; however, some signal was detected in the spleen, lung, and kidney of mice treated with aptamer m12-3773. The combined use of both aptamers significantly ($p < 0.001$) increased the signal in the graft and decreased aptamer accumulation in other tissues. We detected appreciable concentrations of scrambled aptamer only in

the contralateral kidney and in the spleen (Fig. 5a). Because of possible confounding factors related to kidney clearance and PCR artifacts, we repeated the experiments in NOG mice transplanted with human islets in the epididymal fat pad (EFP). Intravenous administration of either aptamer (12.5 pmoles/g) conjugated to streptavidin-AF750 was sufficient to give an appreciable signal over the background in the EFP region as detected by In Vivo Imaging System (IVIS) and, as observed by qRT-PCR, the equimolar mixture of the two aptamers (6.25 pmoles/g each) gave an improved signal (Fig. 5b). As we have previously shown with other streptavidin-conjugated aptamers, we also observed a non-specific signal in the liver region[30]. Next, we evaluated the pharmacokinetics of the aptamer mixture by IVIS in mice transplanted in the EFP. While we detected signal only in the liver in mice treated with scrambled aptamers (Fig. 5c), we observed a rapid accumulation of islet-specific aptamers in the graft region that peaked between 4 and 18 h after administration and then gradually decreased.

We then evaluated whether these two aptamers could be used to quantify transplanted islets. We transplanted NOG mice in EFP with different amounts of human islets (range 0–500 Islet Equivalent, IEQ), waited 3 weeks to allow proper vascularization, and injected the mice intravenously with an equimolar mixture of AF750-conjugated aptamers m12-3773 and 1-717. IVIS, performed 4 h later, revealed a significant correlation between fluorescence signal and the number of transplanted islets (Fig. 5d). We observed only background signals when we injected scrambled aptamers in islets engrafted mice or islet-specific aptamers in non-transplanted NOG mice (Fig. 5d).

We next evaluated the possible toxicity of RNA aptamers in vivo and in vitro on human islets. Briefly, we injected BALB/c mice i.v. q.d. for 5 days with a double dose (25 pmoles/g) of islet-

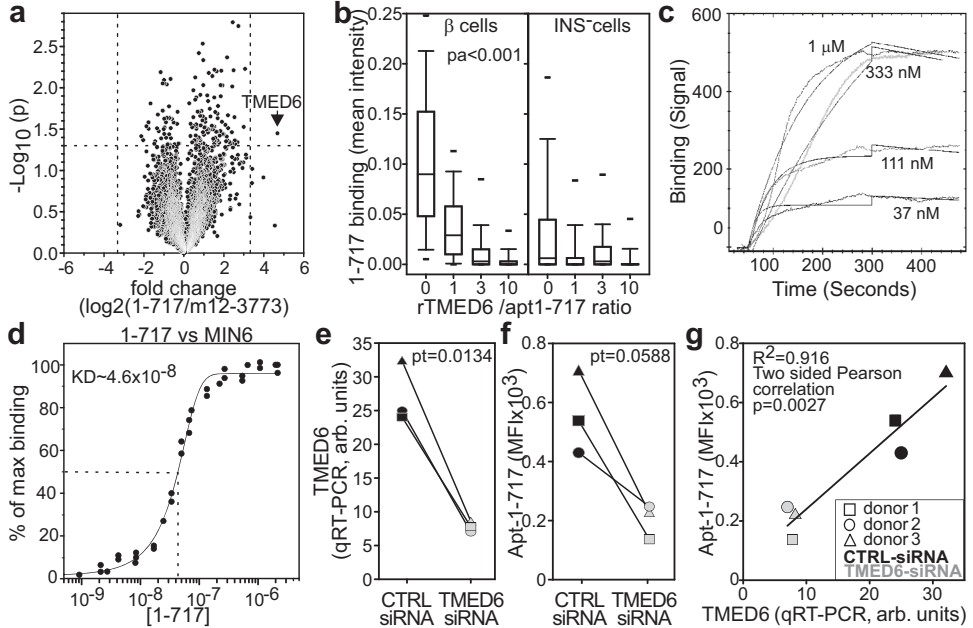

**Fig. 4 TMED6 is the putative target for aptamer 1-717. a** Vulcano plot of aptamer 1-717 over m12-3773 binding on protein arrays Two sides paired *T* test was performed. **b** Cold target inhibition assay. cy3-labeled aptamer 1-717 was admixed at different molar ratios with unlabeled recombinant TMED6. The mixture was then used as a probe against a section of human pancreata. Sections were counterstained with anti-insulin antibody and DAPI; images were acquired and processed with Cell Profiler. Data derived from two independent experiments evaluating 450, 528, 449, and 491 INS+cells and 2590, 3063, 2875, and 2775 cells INS-cells for 0, 1, 3, and 10 rTMED6/aptamer ratio, respectively. The box plot shows the median, the 25, 75, 10, 90, 5, and 95th percentile. One-way ANOVA followed by Holm–Sidak multiple pairwise comparisons was performed. **c** Aptamer 1-717 was run at the indicated concentration as analyte against recombinant TMED6 (ligand) in a surface plasmon resonance assay. **d** Affinity. MIN6 cells were stained with vital dye, and different concentration 1-717 aptamer-chimera annealed with cy5-labeled guide saRNA and analyzed by flow cytometry. **e** Human islets (250 IEQ) were treated with siRNA_TMED6 aptamer m12-3773 chimera or with siRNA_scrambled aptamer m12-3773 chimera and TMED6 expression was assessed by qRT-PCR 72 h later. Two sides paired *T* test was performed. **f** Part of the islets from (**e**) were dissociated and labeled with 1-717 aptamer and anti-insulin antibody and analyzed by flow cytometry. Two sides paired *T* test was performed. **g** Correlation between aptamer 1-717 signal and TMED6 expression. Source data are provided as a Source Data file.

specific aptamers, scrambled aptamers, or PBS. We chose to use immunocompetent BALB/c mice rather than immunodeficient mice to maximize the eventual toxicity mediated by toll-like receptors that may interact with dsRNA despite aptamers' fluorinated backbone. We did not detect any difference in the cellular or chemical parameters evaluated in the complete blood panel performed 2 days after the last injection (Supplementary Fig. 8a).

To detect eventual hampering of islet viability or functionality, we cultured human islets for 24 h with a dose of aptamers ten times higher (2.5 nmoles) than the one given in vivo. Islets were then washed, and viability and function (insulin secretion) were evaluated by the Fluorescein Diacetate/Propidium Iodide Assay and dynamic perifusion, respectively. We did not observe differences in cell viability (Supplementary Fig. 8b) or insulin release to glucose stimuli on islets treated with β cell-specific aptamers compared to scrambled aptamers or PBS (Supplementary Fig. 8c). Taken together, these results strongly suggest that the selected aptamers are safe, can recognize human islets in vivo, and that their combinatorial use provides sufficient sensitivity to discriminate differences of less than 150 IEQ despite the current limitations inherent to IVIS technology[31].

**Aptamer 1-717 and m12-3773 cross-react with mouse islets.** Since aptamer m12-3773 was identified from the mouse-human toggle SELEX and aptamer 1-717 closely clusters with other aptamers identified by toggle SELEX (Fig. 1h), we evaluated their

potential cross-reactivity with mouse islets by immune fluorescence microscopy on pancreatic sections. Both aptamers recognized mouse islets with preferential binding to β cells with only background signal detected in the surrounding acinar tissue (Fig. 5a). Additional analyses using mouse tissue arrays confirmed the specificity of both aptamers for the islets within the pancreas. However, aptamer 1-717 showed binding to the spleen and jejunum, and aptamer m12-3773 to the kidney and the stomach with much lower intensity (Fig. 6b and Supplementary Fig. 9). Considering the data from human tissue arrays (Fig. 2), these findings suggest a differential distribution of the epitopes recognized by aptamers 1-717 and m12-3773 between humans and mice.

To determine whether aptamers 1-717 and m12-3773 could recognize endogenous β cells in vivo, BALB/c mice were injected intravenously with an AF647-conjugated mixture of both aptamers. Mice were euthanized 4 h later, resected pancreases imaged by IVIS (Fig. 6c), snap-frozen, and counterstained in vitro with antibodies against glucagon and insulin (Fig. 6d). While we did not observe any signal in the pancreas of mice treated with the control aptamers, we detected a strong signal in the pancreas from the mice treated with the relevant aptamers (Fig. 6c). Ex vivo immune fluorescence analysis of the pancreatic tissues further revealed specific staining of mouse islets and, in particular, of β cells (Fig. 5d).

Given the ability of the selected aptamers to recognize β cells in vivo, we evaluated whether aptamers 1-717 and m12-3773 could be used to monitor the rejection of islet allografts. Briefly,

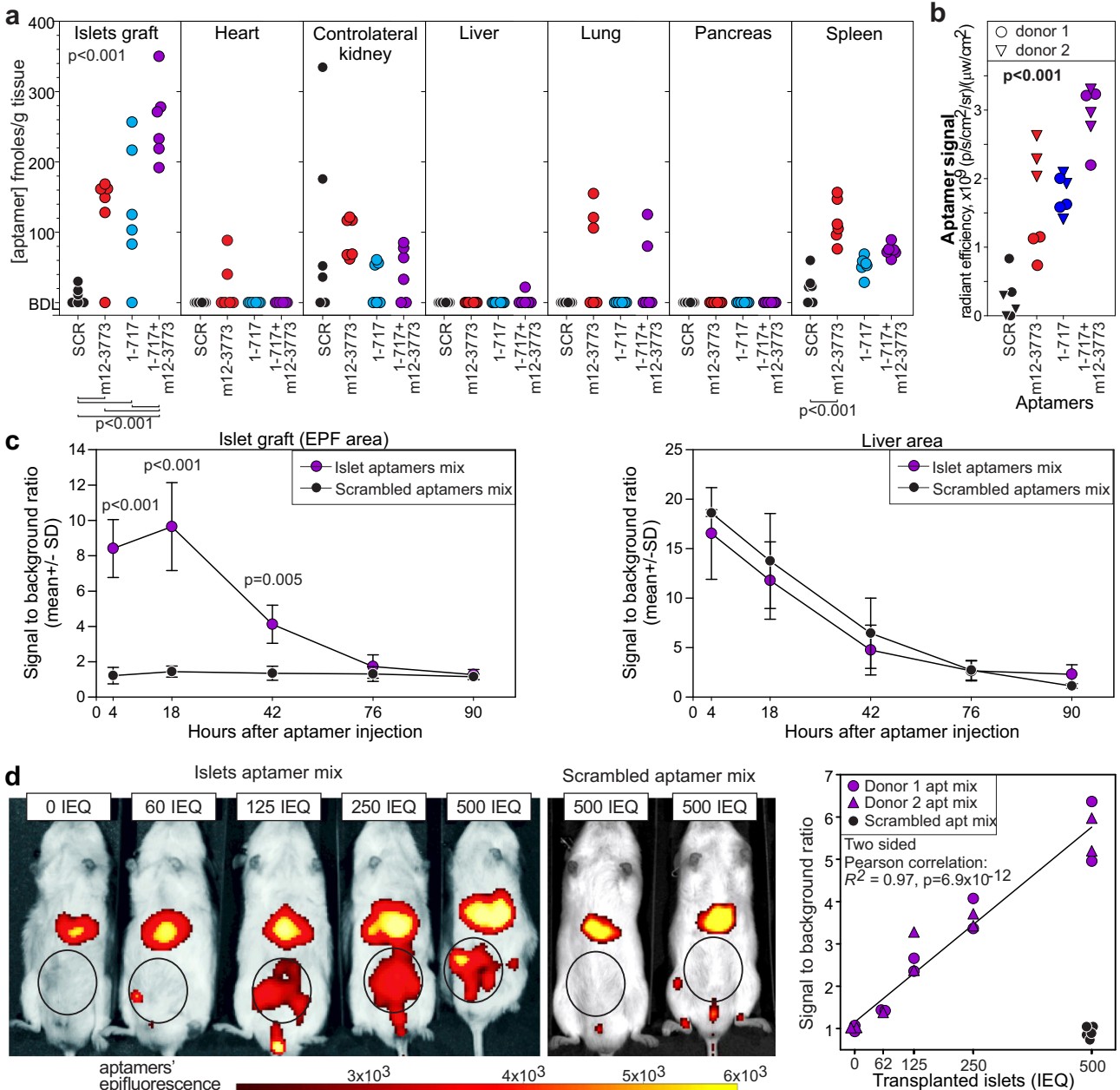

**Fig. 5 Aptamer 1-717 and m12-3773 allow to measure human islet mass in vivo. a** Aptamer 1-717 (12.5 pmoles/g), m12-3773 (12.5 pmoles/g), or an equimolar mixture of the two aptamers (6.25 pmoles/g each) were injected i.v. in NOG mice transplanted 21 days before with human islets (500 IEQ) under the kidney capsule. A mixture of scrambled aptamers was used as a negative control. After 24 h from aptamers injection, organs were collected, and aptamer concentration was determined by qRT-PCR. Two-tailed two-way ANOVA followed by Bonferroni multiple pairwise comparisons was performed. **b** Biotinylated aptamers 1-717, m12-3773, or an equimolar mixture of the two aptamers were complexed with AF750-streptavidin and injected i.v. in immunodeficient NOG mice transplanted in the EFP with human islets (500 IEQ) 21 days before. IVIS was performed 4 h later. The signal from the graft site is reported. Two-tailed one-way ANOVA and posthoc multiple comparison $p$ values are reported. **c** An equimolar mixture (12.5 pmoles/g) of aptamer 1-717 and aptamer m12-3773 complexed to AF750-streptavidin was injected i.v. in NOG mice transplanted 21 days earlier with human islets (500 IEQ) in the EFP IVIS was performed at the indicated timepoints. Data derived from $n = 6$ mice from two independent experiments. Two-sided, three-way ANOVA followed by Holm–Sidak multiple comparison analysis was performed. Multiple comparisons adjusted the $p$ value of islet aptamer vs. scrambled are shown. **d** NOG mice were transplanted with different amounts of human islets in the EFP and injected i.v. 21 days later with AF750-streptavidin aptamer complexes. Signal in the graft site was measured by IVIS 4 h later. Data were derived from two independent experiments performed by two experimentalists. Pearson correlation is shown. Source data are provided as a Source Data file.

BALB/c mice were subcutaneously transplanted with islets from C57BL/6J mice dorsally in the left flank and contralaterally with syngeneic islets as control (Fig. 5e). In vivo islet quantification was performed longitudinally by injecting AF750-conjugated equimolar mixture of aptamers m12-3773 and 1-717 i.v. and by performing IVIS 4 h after each injection. While signals from both grafts were identifiable at the early timepoints, the signal from the allogeneic C57BL6 derived islets was gradually lost. In contrast, the one from the syngeneic graft was retained for the entire observation period (Fig. 6f, g). Graft survival curves revealed that,

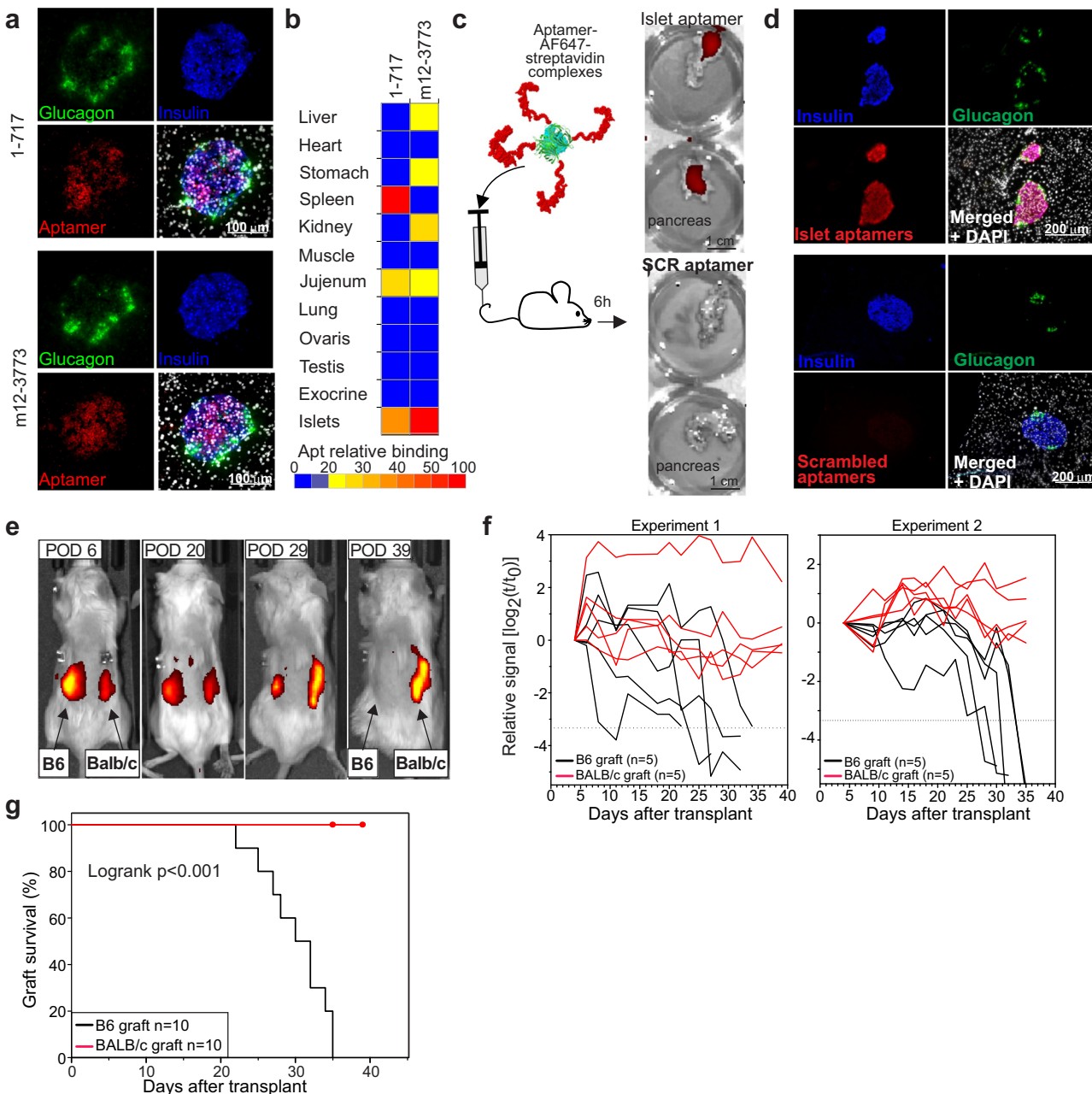

**Fig. 6 Aptamers 1-717 and m12-3773 cross-react with mouse islets in vitro and in vivo. a** Sections of BALB/c pancreatic tissues stained with aptamers 1-717 and m12-3773 and antibodies against insulin and glucagon were imaged with a confocal microscope. A representative image of two independent experiments is shown. **b** Mouse tissue arrays were labeled with aptamers 1-717 and m12-3773, images acquired with a fluorescence microscope, and processed with Cell Profiler **c** An equimolar mixture of biotinylated aptamers 1-717 and m12-3773 or a mixture of the corresponding scrambled aptamers were conjugated with AF647-conjugated streptavidin and injected i.v. in BALB/c mice. Four hours later, pancreases were removed and analyzed by IVIS. A representative image of two independent experiments is shown. **d** Pancreases from (**c**) were snapped frozen, sections cut, and counterstained with anti-insulin and anti-glugagon antibodies. Images were taken with a fluorescence microscope. A representative image of two independent experiments is shown. **e** BALB/c mice were transplanted s.c. dorsally with allogeneic islets from C57BL/6 mice on the left and islets from syngenic islets on the right. An equimolar mixture of aptamers 1-717 and m12-3773 conjugated to AF750-streptavidin was injected intravenously at different timepoints. Four hours after injection, mice were imaged by IVIS. **f** Two independent experiments with $n = 5$ mice each are shown. **g** Survival curve of syngeneic and allogenic grafts. A graft was considered lost when the fluorescence signal was similar to the backgrounds (~10 times lower than the initial signal). Data derived from $n = 10$ mice from two independent experiments. Log-rank test was performed. Source data are provided as a Source Data file.

as expected, all C57BL/6 allogeneic grafts were rejected within 40 days, whereas only one syngeneic (BALB/c) graft was rejected within a week after engraftment. Taken together, these data indicate that aptamers 1-717 and m12-3773 recognize islet cells in vivo and effectively detect islet immune rejection.

**Aptamer 1-717 and m12-3773 conjugated with XIAP specific saRNA increase islet transplantation efficiency.** Transfection of islets with viral vectors encoding for *XIAP*, a gene whose protein inhibits caspase 3, 7, and 9, prevents early graft loss and improves islet survival[32–39]. However, concerns regarding the safety and

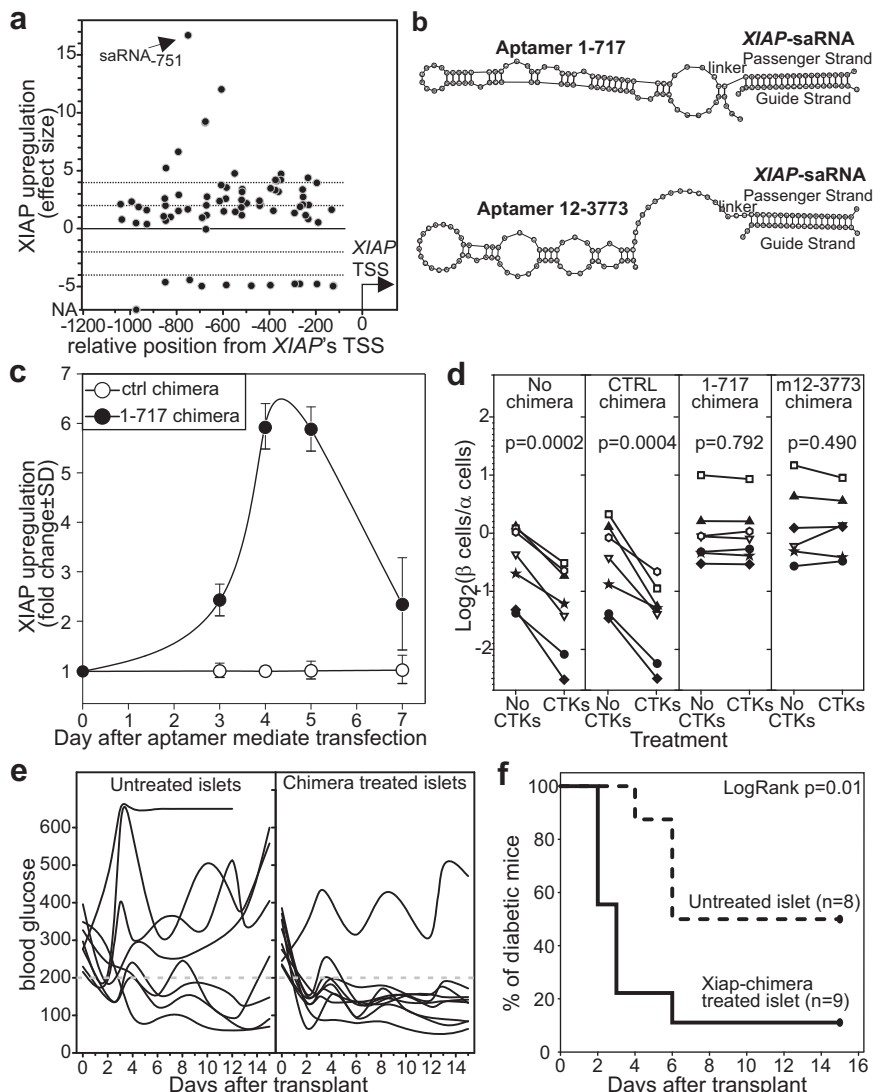

**Fig. 7 Aptamer 1-717 and m12-3773 conjugated with saRNA$_{XIAP}$ prevent early graft loss and improve the efficacy of islet transplantation.**
**a** Identification of saRNAs able to upregulate human XIAP. A549 cells were left untreated or transfected with 75 putative saRNAs identified by bioinformatic analysis on the genomic region between −100 and −1200 bp from the XIAP transcription starting site (TSS). Forty-eight hours later, XIAP expression was evaluated by qRT-PCR. Results are expressed as effect size [$(2^{-\Delta CTexp} - 2^{-\Delta CTctrl})/SD_{exp}$] from three replicates. **b** Chimeras schematic representation. Chimeras were generated by extending the 3′ end of the aptamer with a CC link and the relevant passenger strand and subsequent annealing the desired saRNA. **c** saRNA$_{-751}$/1-717 chimeras or saRNA$_{scrambled}$/1-717 chimeras were added to human islet culture. XIAP expression was evaluated by qRT-PCR at the indicated timepoints. Mean, and standard deviation from $n = 3$ biologically independent samples are shown. **d** Chimeras prevent cytokine-induced β cell loss. Chimeras (420 pmoles) composed of saRNA$_{-751}$ and either aptamer 1-717 or m12-3773 were added to human islets (250 IEQ). Negative controls included untreated cells and cells treated with scrambled saRNA conjugated to either 1-717 or m12-3773 aptamers. Forty-eight hours after transfection, islets were challenged with TNFα, IL-1β, and IFNγ, and the β/α cell ratio was determined by flow cytometry. Paired $t$ test $p$ values are shown. **e, f** Chimeras improve the efficacy of islet transplantation. Human islets (500 IEQ) were treated 24 and 48 h after isolation with an equimolar mixture saRNA$_{-751}$/1-717 and saRNA$_{-751}$/m12-3773 chimeras (420 pmoles each) and immediately transplanted under the kidney capsule of diabetic NOG mice. Untreated islets from the same preparation were used as control. Blood glycemia was monitored three times a week. Data are derived from two independent experiments. **e** Blood glucose concentration of the individual mouse. **f** Kaplan–Meyer curve and log-rank survival analysis. Source data are provided as a Source Data file.

efficiency of the viral approach reduce the possible translation potential of these strategies in clinical settings because of insertional mutagenesis, higher graft immunogenicity, alteration of β cell phenotype, and increased oncogenic risk[40–45]. Here, we explored the possibility of using small activating (sa)RNA, an emerging class of RNA therapeutics that allow to epigenetically upregulate the gene of interest[46–49], in conjunction with the present β cell-specific aptamers to upregulate XIAP in non-dissociated human islet before engraftment. We hypothesized

that this treatment could reduce early graft loss and improve the efficacy of human islet transplantation.

We identified putative saRNAs using the previously described algorithm[50] and selected the 88 target sequences with the highest score for empirical validation (Supplementary Table 6). We transfected human A549 cells, chosen for their low baseline expression level of XIAP[51], with each saRNA and evaluated XIAP expression 96 h later by qRT-PCR (Fig. 7a). Approximately 19% of the tested saRNAs showed a strong and consistent XIAP

upregulation (effect size >4), ~30% a modest upregulation (4 ≥ effect size > 2), and ~15% a significant downregulation (Fig. 7a, Supplementary Table 6). We then conjugated the saRNA yielding the highest effect size (saRNA$_{XIAP-751}$) to islet-specific aptamers (Fig. 7b) and used it to transfect human islets. The simple addition of saRNA$_{XIAP-751}$/aptamer chimera to the islets upregulated XIAP with a maximum peak at days 4–5 and a still significant upregulation at day 7 after transfection (Fig. 7c).

We then tested the saRNA$_{XIAP-751}$/aptamer chimeras functionally on multiple preparations of human islets undergoing challenge with inflammatory cytokines (Fig. 7d). We treated undissociated islets with saRNA$_{Xiap-751}$ conjugated to aptamer 1-717 or m12-3773 or with scrambled saRNA-aptamer chimera as control. Untreated islets were used as an additional control. Two days after transfection, we challenged the islets with inflammatory cytokines to induce β cell death. One day after cytokine addition, islets were dissociated and analyzed by flow cytometry (Fig. 7c). As expected, the inflammatory cytokines significantly reduced the β to α cell ratio in the islets left untreated or treated with the saRNA$_{scrambled}$/aptamer-chimera indicating β cell death. Conversely, we observed no decrease in the β to α cell ratio in the groups treated with saRNA$_{XIAP-751}$ aptamer chimera (Fig. 7d).

Although exposure to inflammatory cytokines can serve as in vitro surrogate of early graft loss, additional factors (e.g., hypoxia, hyperglycemia) can affect β cell viability upon engraftment. We thus performed marginal mass human islet transplantation experiments in immune deficient NOG mice to evaluate the efficacy of saRNA$_{XIAP}$-chimeras as islet treatment before engraftment. We treated human islets (500 IEQ) with saRNA$_{XIAP-751}$ conjugated to a mixture of aptamers 1-717 and m12-3773. The rationale for the simultaneous use of both aptamers was: (i) the improved effect that the combinatorial use of both aptamers provided (Fig. 5a) and (ii) the need to minimize the eventual effect of the genetic polymorphisms in *TMED6* and *clusterin* among donors[52,53]. We then engrafted treated or untreated islets under the kidney capsule of streptozotocin (STZ)-treated diabetic NOG mice. Blood glycemia was used for the functional assessment (read-out) of islet transplantation (Fig. 7e). Approximately 50% of the mice engrafted with untreated islets became euglycemic in this setting. Conversely, almost all the mice transplanted with islets treated with saRNA$_{XIAP-751}$ aptamer chimeras were cured (Fig. 7f). Notably, while the control group reached euglycemia by day 6, mice receiving islets treated with chimeras reverted diabetes by day 2 (Fig. 7e). Taken together, these data highlight the potential of our aptamers as transfection reagents for β cells in the islets and provide an important tool to significantly improve the efficiency of human islet transplantation by decreasing early graft loss.

## Discussion

The identification of β cell-specific epitopes and their cognate ligands represents an important medical need to better understand disease progression before clinical manifestation in patients at risk of diabetes. Such β cell-specific probes would also help rapidly screen new therapeutic approaches and pave the way for new targeted therapies[54]. Most strategies pursued toward this goal have been based either on differential transcriptomics or proteomic approaches[55–57]. The first strategy does not account for the posttranslational modification of membrane proteins and differential intracellular localization of targets within cell types. The second one is still hindered by technical limitations in membrane protein isolation, mass spectrometry analysis algorithms, and difficulty discriminating proteins differentially expressed among cell types[58–61]. Additionally, if a target is identified by proteomics or transcriptomics, considerable effort is required to generate drugs that bind specifically to the identified molecule(s).

Here, we explored an unsupervised approach based on the selection of RNA aptamers that identify targets differentially expressed between acinar tissue and pancreatic islets in mice and humans. This screening strategy has the potential to directly identify ligands against intact targets differentially and natively expressed in different cell types. We optimized and used Cell SELEX on a cluster of cells (islets) instead of a single-cell suspension to minimize artifacts and/or damage to membrane proteins caused by dissociating agents. We employed both a "cluster-cell" SELEX using only human specimens and a "toggle" SELEX in which selection was first performed with mouse islets and then toggled to human islets and acinar tissue (Fig. 1). Bioinformatic analyses on the polyclonal libraries from both unsupervised methods revealed a convergent selection towards monoclonal clones with a putative specificity against human islets (Fig. 1h). Indeed, 12 of the 15 monoclonal aptamers selected for empirical validation showed higher binding to human islets over the surrounding exocrine tissue (Fig. 2a). Since acinar tissue does not represent the full complexity of the proteins present in other tissues, we employed FDA-approved tissue arrays to identify the candidates with superior specificity. This screening highlighted two lead candidates (aptamers 1-717 and m12-3773) that showed excellent specificity to human islets and low binding to the 29 other tissues evaluated (Fig. 2b and Supplementary Fig. 2). Interestingly, a different binding distribution among the non-pancreatic tissue was observed in mouse tissue arrays (Fig. 6b and Supplementary Fig. 9), suggesting a differential biodistribution of the two cognate targets in mice and humans. Intra-pancreatic specificity, as assessed by image cytometry using fluorescence and confocal microscopy and flow cytometry, highlighted a preferential binding of both aptamers to β cells. However, aptamer m12-3773 and 1-717 showed some binding at low intensity to a few insulin negative cells within the endocrine and exocrine pancreatic tissue (Fig. 2). It is important to note that both aptamer 1-717 and m12-3773 recognize different β cells at different intensities suggesting a differential expression of their targets within β cell subsets. This suggests the combinatorial use of multiple aptamers, each specific for a different β cell-specific epitope, to maximize signal to background ratio. The combinatorial use of multiple β cell-specific aptamers should maximize the number of targeted insulin-producing cells, increase the quantity of imaging reagent delivered to the cells expressing both targets, and minimize binding to other tissues and cells expressing only one target.

Both aptamers recognized mouse and human islets in vitro and, more importantly, in vivo (Figs. 5, 6). Our in vivo biodistribution experiments (Fig. 5) sustain the concept of combinatorial specificity since the signal observed with an equimolar mixture of the two aptamers significantly increased the specific signal in the islet graft and lowered the non-specific signal in other tissues. Indeed, the binding of β cells targeting agents to non-insulin-producing cells is a problem also reported with other probes. For example, antibodies against GLP1R recognize the pancreatic ducts and other cells in the exocrine pancreas and the thyroid[62]. Similarly, VMAT2 expression is not entirely restricted to β-cells but is also expressed in a fraction of pancreatic polypeptide secreting cells and on pancreatic nerves[63]. The combinatorial use of multiple imaging probes, each specific for a marker preferentially (although not exclusively) expressed by β cells, has the potential to increase our capacity to measure β cells accurately. Indeed, the systemic administration of an equimolar mixture aptamers 1-717 and m12-3773 yielded a fluorescence signal in the graft region proportional to the number of transplanted islets (Fig. 5), indicating that the combinatorial use of our

aptamers can be used to measure β cell mass. Similar experiments performed in BALB/c mice showed that a mixture of both aptamers could efficiently label the endogenous β cells in vivo and monitor the rejection of islet allografts (Fig. 6).

Clusterin and TMED6 have been identified as putative targets for aptamers 1-717 and m12-3773, respectively (Figs. 3, 4). Interestingly, two different techniques (immunoprecipitation followed by mass spectrometry and aptamer labeling of high-density protein arrays) had to be used for these identifications, likely due to the intrinsic nature of these two targets and possible clusterin posttranslational modifications.

SPR and flow cytometry assays showed different affinities of our aptamers for the cognate targets. While SPR and flow cytometry against MIN6 cells indicate a KD in the nanomolar range, flow cytometry against primary human β cells gave an apparent KD of $\sim 1 \times 10^{-7}$ that might not suffice for PET and SPECT probes. Although the inevitable presence of dead cells after islet dissociation can explain this suboptimal affinity for primary human β cells[64], the future imaging probe might require improvements using, for example, multimeric forms of our aptamers.

Human clusterin is a multifunctional glycoprotein with multiple isoforms (6 isolated proteoforms and ten from transcriptome studies) that interacts with lipids, amyloid proteins, complement components, immunoglobulins, and misfolded proteins[65]. Clusterin is ubiquitously expressed but with variable molecular weights and possibly different glycosylation patterns depending on tissues and cell types[66–68]. The various clusterin proteoforms are functionally distinct and have been found in the nuclei, cytoplasm, or plasma membrane or as being secreted[69–73]. This high variability of clusterin can explain the various functions reported for different tissues and cell types[72,73]. Thus, despite the ubiquitous presence of clusterin, we cannot exclude the presence of transcriptional, translational, and/or posttranslational modification of clusterin in β cells resulting in specific epitopes. Clusterin is highly expressed in human islets[74] and plays a key role during pancreas development in the fetus, in β cells regeneration after partial pancreatectomy, and in the differentiation of duct cells into insulin-producing cells[75,76]. Its presence in the blood has been suggested as a biomarker for T1D[55]. Clusterin can bind insulin, and its deficiency exacerbates insulin resistance in T2D[77]. These data suggest a possible active role of this chaperon protein in allowing correct insulin folding and secretion.

All the peptide fragments identified by mass spectrometry after aptamer m12-3773 based immunoprecipitation reside in the alpha chain and the isoform 3 of clusterin (Fig. 3). However, we still do not know the exact role of clusterin in islet cells and which isoforms or posttranslational modifications are present in β cells; these will be part of future studies that will also serve to further validate this protein as a target for delivering imaging reagents and therapeutics to β cell in human. Posttranslational modification of clusterin may also explain why aptamer m12-3773 did not reliably bind to different commercially available recombinant human clusterin or protein arrays but did bind to clusterin in islets as suggested by our immunoprecipitation and mass spectrometry data and silencing experiments (Fig. 3).

TMED6 is a poorly studied type 1 transmembrane protein that seems to play a regulatory role in vesicle assembling and trafficking[78,79]. Expressed tag analysis indicates that this gene is overexpressed in the pancreas compared to other tissues at a level just below those of glucagon and insulin[80]. Further studies suggest that transcripts for this protein are upregulated in islet cells, and its silencing drastically reduces insulin secretion in MIN6 cells[80]. This suggests a potential role of this protein in controlling insulin vesicles[80]. Indeed, a possible role of TMED6 in insulin secretion is also indicated by the correlation of TMED6

downregulation in islets during the development of T2D in Goto-Kakizaki rats[80] and by a polymorphic association of the region of this gene with T2D[81]. It is important to note that most of the studies on TMED6 in the pancreas relied on poorly validated polyclonal antibodies and gave contradictory results with staining either in the acinar tissue (https://www.proteinatlas.org/ENSG00000157315-TMED6/tissue/pancreas#img) or specifically in β cells[80]. The specificity of aptamer 1-717 is supported by binding to the recombinant protein (Fig. 3e), binding to protein array, cold target inhibition assays (Fig. 4b), SPR analysis (Fig. 3c), and aptamer-mediated silencing experiments (Fig. 4e–g). Thus, aptamer 1-717 seems the only validated reagent currently available to study this protein.

Validation using aptamer-siRNA chimera also highlighted aptamers 1-717 and m12-3773 as highly efficient transfection reagents that modulate the expression of the desired gene in non-dissociated islets (Figs. 3b and 4e). We employed this property therapeutically in a clinically relevant marginal mass transplantation setting. The simple addition of picomolar quantities of saRNA$_{XIAP}$ aptamer chimeras to non-dissociated human islet preparation was sufficient to upregulate XIAP for up to 7 days, prevent cytokine-induced β cell death, and, most importantly, significantly increase the efficacy of islet transplantation (Fig. 7). In particular, aptamer-mediated XIAaptamer-mediateddeduced the number of islets needed to reverse diabetes by more than 50%, a gain that is in line with the one reported with the use of adenoviral vectors[35] and that is unprecedented for non-viral vectors. It is important to remember that early graft loss seriously compromises the efficacy of both autologous and allogeneic islet transplantation. In patients undergoing total pancreatectomy with islet autotransplantation because of chronic pancreatitis or injury, early graft loss is one of the leading causes for the poor efficiency of this FDA-approved procedure, which prevented iatrogenic diabetes only in approximately 30% of the patients[82–84]. The use of saRNA$_{xiap}$-aptamer chimera on islets just before transplantation may significantly improve the outcome for these patients by reducing early graft loss. Similarly, T1D patients with untreatable and impaired awareness of hypoglycemia and history of severe hyperglycemic episodes are treated with islet transplantations from multiple donors to offset the massive loss of viable β cells soon after transplantation[85–88]. The use of our aptamer-saRNA$_{XIAP}$-chimera could open a realistic possibility to improve the outcome of islet transplantation by inhibiting early graft loss and thus reducing the number of donors needed and the risk associated with recipient sensitization toward multiple allogeneic human leukocyte antigens (HLA)[89,90]

In summary, we reported the identification of RNA aptamers specific for TMED6 and clusterin that show highly selective binding to human and mouse β cells in vitro and in vivo. These aptamers can be easily conjugated with imaging reagents for β cell mass quantification and RNA therapeutics for the efficient non-viral transfection of human β cells. Our results also suggest the combinatorial use of multiple aptamers or probes with different specificity to improve our capacity to measure β cell mass or deliver therapeutic cargo to insulin-producing cells in vitro and in vivo.

## Methods
All key resources are summarized in Table 1.

**Ethics approval**. All animal experiments were performed according to all relevant ethical regulations and were approved by the Division of Veterinary Resources and the Institutional Animal Care and Use Committee of the University of Miami IAUC protocol 19-068.

Human islets and samples from male and female cadaveric donors were acquired through commercial vendors and/or tissues banks with no information linking them to the donors. According to regulation 45 CFR 46.of the U.S. Department of health and human services (https://www.hhs.gov) defining a human

**Table 1 List of key resources used in the study.**

| Reagents and resources | Source | Identifier |
|---|---|---|
| **Antibodies and aptamers** | | |
| Guinea pig anti-Insulin (1:300) | DAKO | Cat# A0564; RRID: AB_2617169 |
| Rabbit anti-Glucagon (1:300) | Cell Signaling | Cat# 2760; RRID: AB_659831 |
| Anti-Insulin APC (1:10) | R&D Systems | Cat# IC1417A, RRID:AB_2126535 |
| Anti-glucagon PB (1:100) | BD Biosciences | Cat# 565860, RRID: AB_2739382 |
| Goat anti-Rabbit IgG, AF 488 (1:400) | Thermo Fisher Scientific | Cat# A-11034, RRID:AB_2576217 |
| Goat anti-GuineaPig IgG AF647 (1:400) | Thermo Fisher Scientific | Cat# A-21450, RRID:AB_2735091 |
| 5'Biotin-1-717, 5'Biotin-m12-3773, 5'Biotin-SCR-1-717, 5'biotin-SCR-M12-3773 | Oligofactory | Custom order (Supplementary Table 3) |
| **Biological samples** | | |
| Human pancreatic islets | Diabetes Research Institute, Miami | |
| Human pancreatic islets | Prodo labs | https://prodolabs.com |
| Mouse pancreatic islets | Animal Core (DRI) | NA |
| A549 cells | ATCC | Cat#CRM-CCL-185, |
| **Chemicals** | | |
| L/D yellow | Invitrogen | Cat#: L34968 |
| DAPI (1 μg/ml) | Invitrogen | Cat#: D1306 |
| Dextran sulfate sodium salt (1:2) | Pharmacia Biotech | Cat#: 17-0340-01 |
| Fixation/permeabilization solution kit | BD Cytofix/Cytoperm™ | Cat#: 554714 |
| Streptavidin, AF750 | Life technologies | Cat#: S21384 |
| Streptavidin, AF647 | Biolegend | Cat#: 405237 |
| Silencer™ siRNA labeling kit Cy™3 | Ambion | Cat#: AM1632 |
| Albumin solution from bovine serum | Sigma | Cat#: A9576 |
| GlycoBlue 300 μl, 15 mg/ml | Ambion | Cat#: AM9515 |
| 2-Propanol | Sigma | Cat#: I9616 |
| 3M sodium acetate (pH5.5) | Ambion | Cat#: AM9740 |
| Nuclease free water | Teknova | Cat#: W3440 |
| Taq DNA polymerase | Invitrogen | Cat#: 10342053 |
| DuraScribe® T7 transcription Kit | Lucigen | Cat#: DS010925 |
| SuperScript III reverse transcriptase | Invitrogen | Cat#: 18080085 |
| Dynabeads™MyOne™ Streptavidin C1 | Invitrogen | Cat#: 65001 |
| 0.25%Trypsin-EDTA | Gibco | Cat#: 25200-056 |
| Pen Strep | Gibco | Cat#: 15140-122 |
| RPMI 1640 medium | Gibco | Cat#: 11875119 |
| Islet media PIM (R) (5.8 mM glucose) | PRODO | Cat#: PIM-R001GMP |
| FBS | Gibco | Cat#: 16000-044 |
| PBS | Gibco | Cat#: 10010-023 |
| TRIZOL | Ambion | Cat#: 15596018 |
| High-capacity cDNA RT kit | Applied Biosystems | Cat#: 4368813 |
| TaqMan universal PCR master mix | Applied Biosystems | Cat#: 4352042 |
| EUK 18S rRNA (DQ) Oligo mix | Applied Biosystems | Cat#: 4352655 |
| TMED6 gene expression assay | Applied Biosystems | Cat#:4400291/Hs00376251_m1 |
| CLU gene expression assay | Applied Biosystems | Cat#:4400291/Hs00156548_m1 |
| XIAP gene expression assay | Applied Biosystems | Cat#: 4400291/Hs04107956_cn |
| rhTMED6 | Abcam | Cat#: ab165536 |
| Human TNF-α | Peprotech | Cat#: 300-01A |
| Human IL-1ß | Peprotech | Cat#: 200-01B |
| Human IFN-γ | Peprotech | Cat#: 300-02 |
| Lipofectamine 3000 | Invitrogen | Cat#: L3000008 |
| **Critical commercial assays** | | |
| RNeasy Mini kit | QIAGEN | Cat#: 74104 |
| QIAquick PCR purification kit | QIAGEN | Cat#: 28106 |
| Amicon Aultra-4 50k | Millipore | Cat#: UFC805024 |
| Arrayit_HuProt™v2.0-19K-Human Proteome Microarray | ArrayIT | Cat#: HP19K |
| BlockIT buffer | ArrayIT | Cat#: BKT |
| Chemblock Microarray Blocking buffer | ArrayIT | Cat#: CHE |
| FDA Standard Frozen Tissue Array—Human Adult Normal | Biochain | Cat#: T6234701-2 |
| Ready-to-Use Mouse Mixed Frozen Tissue Microarray, 2 | AMSBIO | Cat#: MAF-MT2 |
| **Experimental models: organisms and strains** | | |
| NOD.Cg-Prkdcscid Il2rgtm1Sug/JicTac | Taconic | https://www.taconic.com/mouse-model/ciea-nog-mouse |
| BALB/c | Jackson Labs | https://www.jax.org/strain/000651 |
| C57BL/6J | Jackson Labs | https://www.jax.org/strain/000664 |
| **Software and algorithms** | | |
| APTANI | (Caroli et al.[26]) | http://aptani.unimore.it/ |
| FCS6 express Plus | Denovo | https://www.denovosoftware.com/ |

**Table 1 (continued)**

| Reagents and resources | Source | Identifier |
|---|---|---|
| Cell profiler | Cell profiler | https://cellprofiler.org/ |
| Clustal omega | Clustal-Ω | http://www.clustal.org/ |
| saRNA screening algorithm | (Wang et al.[50]) | NA |
| Blat | | http://genome.ucsc.edu/cgi-bin/hgBlat |
| Blast | | https://blast.ncbi.nlm.nih.gov, |
| siRNA wizard | | Invivogen https://www.invivogen.com/sirnawizard/siRNA.php |

Further information and requests for resources and reagents should be directed to and fulfilled by the Lead Contact, P.S. pserafini@miami.edu.

subject as "a living individual about whom an investigator (whether professional or student) conducting research" with emphasis given on the living, any research involving cadavers, autopsy material, or biospecimens from now deceased individuals does not meet the regulatory definition of "human subject research" and since no protected health information were used in the study, no IRB oversight nor approval was required for this study.

**HT-"Cluster-cell" SELEX and Toggle SELEX**. The random DNA library (Supplementary Table 3), previously described by the Sullenger group[91], was amplified by PCR using recombinant Taq (Invitrogen) with the Sul5′ and the Sul3′-short primers (Supplementary Table 3) with the following cycling condition: 95 °C 5′, 3× (94 °C 30″, 52 °C 20″, 72 °C 25″), 20× (94 °C 30″, 54 °C 20″, 72 °C 25″), 72 °C 5′. The amplicons were purified using the PCR purification kit (Qiagen) and transcribed in vitro by Durascribe T7 RNA synthesis kit (Lucigen, USA). The resulting 2′Fluoro-RNA aptamers were purified using the RNeasy kit (Qiagen). In the initial selection round, 200 pmoles of RNA were suspended in 450 μl PBS (pH 7.4, Life technologies, MO, USA), heated at 65 °C for 5′ and then cooled at room temperature (25 °C—RT) for 10′. In each cycle of selection, the RNA aptamer library (200 picomoles in cycles 1–2 and 100 picomoles in subsequent cycles) derived from the initial random library or the previous selection cycle (cycles 2–8) was first depleted of non-specific aptamers by a 15′ incubation at RT in 1 ml of PBS supplemented with yeast RNA (10 μg/ml, Ambion) on a rotator with dissociated exocrine tissue (~$10^6$ cells) previously depleted from islets by hand. The suspension was spun down (10′ @300 × $g$ 4 °C), and the supernatant passed through a 0.2 μm PES filter. The filtered supernatant containing the unbound RNA aptamers was then used to resuspend handpicked islets (~200 IEQ in cycles 1–4 and ~100 IEQ in cycles 5–8) and incubated for 10′–20′ in rotation at RT. The preparation was then washed 3–6 times with 1 ml of PBS and yeast RNA (10 μg/ml) and centrifuged (10′, @300 × $g$, 4 °C) and cell pellet lysed with Trizol (Thermo fisher Scientific). Recovered RNA aptamers were then cleaned with RNAeasy and reverse transcribed using Sul3′-short primer and the SuperScript® III Reverse Transcriptase. In the HT "cluster-cell" SELEX, we performed eight selection cycles using islet and acinar tissues from 4 different cadaveric donors. Stringency was gradually increased by: (a) limiting the quantity of RNA aptamer used for the selection to 100 picomoles starting from cycle 3, (b) reducing the number of islets to 100 IEQ in cycle 5–8, (c) reducing the incubation time with the islets to 10′ from cycle 6, and by increasing the number of washes after the positive selection as follow: 3 in cycle 1–3, 4 in cycles 5 and 6, and 6 in cycle 7 and 8. Additionally, to minimize PCR artifacts, PCR cycles were reduced to 15 cycles from cycles 4–8. A similar strategy was followed with the toggle SELEX using islets and acinar tissues from BALB/c mice in the first eight cycles (Cycles M1–M8) and islets and acinar tissue from one cadaveric donor in the last two cycles (Cycles M8H1 and M8H2).

**Preparation of libraries for HT-sequencing**. cDNA from cycles 1–8 in the HT "cluster-cell" and cycles M8, M8H1, and M8H2 from the toggle cell SELEX were sampled for HT-sequencing by tagging aptamers' constant region with two sequential PCRs. The first PCR reactions were performed in 100 μl of water containing 1X PCR buffer, MgCl₂ solution (1.5 mM), dNTPs (200 μM each), DNA template (5 ng/μl), recombinant Taq polymerase (5 U, Invitrogen), and the PFA and PRA primers (Supplementary Table 3) corresponding to each cycle described above. The reactions were performed in the GS482 thermocycler (G-STORM) using the following program: 95 °C 5′, 5× (95 °C 1′, 56 °C 30″, 72 °C 30″), 72 °C 10′. PCR was purified via gel extraction using the QIAquick Gel Extraction Kit (QIAGEN) following manufacturer instructions. The second PCR was performed using the same condition described above but using the UFB and PRB primers (Supplementary Table 3) and the following cycles: 95 °C 5′, 6× (95 °C 30″, 65 °C 30″, 72 °C 30″, 72 °C 10′. Products were purified by gel extraction; quality was evaluated via bio-analyzer (Agilent). Library quantitation and pooling took place at the Hussman Institute for Human Genomics-Center for Genome Technology using the KAPA Library Quantification Kit for Illumina platforms (part# KK4854). 10-13 pM of pooled samples were loaded on the Illumina cBot for cluster generation according to the manufacturer's recommendations. Sequencing was performed on an Illumina HiSeq 2000/2500 (HCS 2.0.12.0) using the reagents provided in the Illumina TruSeq PE Cluster Kit v3 and the TruSeq SBS Kit-HS (200 cycles) kit. Data processing was done using HiSeq's Real-Time Analysis (RTA)

from Casava software. Base-calling files were transformed into zipped FASTQ files containing raw reads with base qualities. These raw read files were then filtered by Illumina's internal filter resulting in 2 FASTQ files (1 per read) containing all pass-filter reads. FASTQ files were used as input for frequency evaluation and for Clustal-Ω and APTANI analyses[25,26]. HT-sequencing raw and processed data from SELEX experiments are available in GEO as GSE197262.

**Bioinformatic analysis of aptamer libraries**. FastQ files were processed to identify the different aptamers. Briefly, the aptamers' variable region (40–44 nucleotides) was determined using the surrounding constant regions in 5′ and 3′. Clustering (Clustal-Ω v 1.1.1.2013.05.31) was performed on cycle 8 of human "cluster-Cell" SELEX on the 7158 sequences whose frequency was higher than $10^{-6}$, resulting in 208 families of which only 31 (Supplementary Table 1) had a cumulative frequency (i.e., the sum of all individual sequence frequency within the family) higher than $10^{-4}$ and accounting for 66.7% of the sequences in the library. The most frequent aptamer in each of these 31 families was selected and compared by Clustal Ω with those from the toggle SELEX.

Thirty-nine aptamers from toggle-SELEX (Supplementary Table 2) were selected by: (i) selecting those that were undetected in the M8 library and had a frequency higher than $10^{-4}$ in MH2, (ii) filtering those whose frequency was higher than $5 \times 10^{-6}$ in the M8 library, higher than $10^6$ in MH2, and expand more than ten times after the human selections, and (iii) choosing the one with a frequency higher than $10^4$ in M8 and at least doubling in frequency after the human selection.

Clustal Ω analyzed the resulting 80 aptamers from the two selection strategies, and 15 of those were chosen considering the relative similarity in the cluster analysis and convergent selection using the two different selection strategies as representative for each cluster branch for empirical validation.

Library complexity among cycles was calculated as 100× (number of unique sequences)/(total reads). PCA analysis was performed using Jalview with the "DNA" default parameters[92] on the top 5000 sequences aligned with Clustal Ω.

**Tissue staining**

*Cy3 aptamers*. Aptamers were labeled with cy3 using the Silencer siRNA Labeling Kit—cy3 (Ambion). Fresh frozen tissues and tissue arrays (AMSBIO) were fixed in 10% neutral-buffered formalin (BDH) for 15′ at RT, incubated with dextran sulfate sodium/PBS (1:2 m/V—Pharmacia Biotech) for 30′ and washed with PBS. Then, tissues were stained with cy3-labeled aptamer (10 μg/ml) in PBS for 30′. In the cold target inhibition experiments, the putative protein was added at the indicated molar ratio before staining to cy3-labeled aptamers (60 nM). The aptamer-stained slides were masked with blocking buffer (2% BSA and 10% FBS in PBS) and counterstained with anti-glucagon (Cell Signaling- 1:300 dilution) and anti-insulin (DAKO 1:300) primary antibodies overnight at 4 °C. The slides were washed three times with PBS for 10′ at RT, stained with AlexaFluor-647 anti-guinea pig and AlexaFluor-488 anti-rabbit secondary antibodies for 90′ at RT, washed three times (10′ at RT in PBS), and counterstained with DAPI (10′ at RT followed by two washes with PBS). Sections were imaged using a fluorescence microscope (Zeiss Apotome or Keyence BZ-X800), the Hi-throughput VS120 Olympus slide scanner, or the Leica SP5 confocal microscope.

*Biotinylated aptamers*. Frozen tissues were fixed for 20′ in neutral-buffered formalin solution, washed three times with PBS, treated with Image-IT (Invitrogen) for 30′ to remove background fluorescence, washed with PBS, and blocked with superblock (Invitrogen)-BSA3%-Tween 20 0.05% for 1 h at RT. Specimen were rinsed in PBS, and endogenous biotins were blocked using the endogenous biotin blocking kit (Invitrogen). Tissues were dextran sulfate sodium/PBS (1:2 m/V— Pharmacia Biotech) for 30′ at RT, rinsed with PBS, and incubated with biotinylated aptamers (0.5 μM) in the presence of yeast RNA (Ambion) 1 mg/ml for 30′ at RT. Tissues were washed four times with PBS (5′ RT) and counterstained with guinea-pig anti-human insulin (4C O/N). Tissues were washed four times with PBS Tween 20 (10′ wash at RT), rinsed with PBS, and stained for 1 h at RT with AF594-goat anti-guinea-pig antibody, AF488 mouse anti-human glucagon, and AF647-streptavidin. Sections were washed three times with PBS-tween 20 (0.5%), once in PBS, mounted with Fluorogel + DAPI with the coverslip, and imaged on Keyence fluorescence microscopy.

**Cell profiler**. Tiff files from whole slide scan or individual pictures were converted in grayscale and optimized with ImageJ using CLAHE[93] to reduce eventual illumination artifacts using for DAPI block size 200 and maximum slope 3.5, and for insulin, glucagon, and aptamers channel block size of 20 and maximum slope of 1.1. Images were then sliced into 600 × 600 pixel images using ImageSlicer (https://www.coolutils.com/TotalImageSlicer), each identifiable as metadata for column and row. Sliced images were loaded in the cell-profiler software. Nuclei were identified as the primary object using the blue (DAPI) channel by setting the diameter of the nuclei between 2 and 10 pixels, using the three classes Otsu Adaptive threshold method with a correction factor of 1, and the lower and upper bounds on threshold 0.1–1.0. Clumped objects were distinguished by shape, the size of the smoothing filter and the minimum allowed distance between local maxima were automatically calculated. Secondary objects (i.e., cells) were identified using the autofluorescence and fluorescence of the merged image from the channels acquired using the nuclei propagation method with three classes Otzu Adaptive threshold method, 0.9 as threshold correction factor (0.0–1.0 range) and 0.02 as regularization factor. The cytoplasm was the tertiary object was identified as the area included in the cells (secondary object) but not in the nuclei (primary object). For each cell, the integrated intensity means of the channels in the nuclei, cells, or cytoplasm were exported as a "cpout" file. Cpout files were analyzed using FCS Express 7 PLUS. Examples of segmentation and analysis are provided in Supplementary Fig. 5.

**Cell culture**. Human islets and acinar tissues were isolated by the human islets GMP isolation core at the Diabetes Research Institute maintained in PIM (R) media (PRODO) and used between 24 and 48 h after isolation. Only islets with more than 90% viability and purity have been used for selection. Mouse islets and exocrine tissues were purified by the small animal core at the Diabetes Research Institute from 5 to 6 pancreas of BALB/c mice. Islets and acinar tissues were maintained in PIM (R) media (5.8 mM glucose) and used within 48 h from isolation. A549 cells, a human lung cancer cell line isolated from a male patient[94], were acquired by ATCC and maintained in RPMI 1640 media (10% FBS, 1% Pen/Strep—Invitrogen Thermo Fisher) in 5% $CO_2$ at 37 °C.

**In vitro islet assays**. Islets (500 IEQ) were cultured in 24-well plates in 1 ml of PIM (R) media (5.8 mM glucose) with RNA aptamers (2.5 nMoles) or scrambled aptamer for 24 h at 37C 5%CO2. Islets were then washed and assessed for viability function. Islets viability was assessed following the recommendation of the IDT consortium using the Fluorescein Diacetate/ Propidium Iodide (FDA)/(PI) Viability Assay. 20–30 fields with ×4 magnification each containing 5–20 islets were acquired using a "revolve microscope" and images were analyzed with ImageJ to determine "red" and "green" area and calculate the viability index as follow: Viability index = (green area)/(green area + red area) × 100.

Islet functionality was evaluated by perifusion (dynamic GSIS) experiments as previously described[95]. Briefly, we used a PERI4-02 machine (Biorep Technologies, Miami, FL, USA) that allows parallel perifusion for up to six independent channels. For each experiment, 100 human IEQ were handpicked and loaded in Perspex microcolumns between two layers of acrylamide-based microbead slurry (Bio-Gel P-4, Bio-Rad Laboratories, Hercules, CA, USA). Perifusion buffer containing 125 mM NaCl, 5.9 mM KCl, 1.28 mM CaCl₂, 1.2 mM MgCl₂, 25 mM HEPES, and 0.1% BSA at 37 °C with selected glucose or KCl (25 mM) concentrations was circulated through the columns at a rate of 100 µL/min. After 60 min of washing with low glucose (3 mM) solution for stabilization, islets were stimulated with the following sequence: 8 min of low (3 mM) glucose, 20 min of high (11 mM) glucose, 15 min of low glucose, 10 min of KCl (25 mM), and 10 min of low glucose (3 mM). Samples (100 µL) were collected every minute from the outflow tubing of the columns in an automatic fraction collector designed for a multi-well plate format. Islets and the perifusion solutions were kept at 37 °C in a built-in temperature-controlled chamber while the perifusate in the collecting plate was kept at <4 °C to preserve the integrity of the analytes. Insulin concentrations were determined by ELISA (Mercodia Inc., Winston Salem, NC, USA). Data were normalized on the DNA contents (evaluated using the dsDNA Picogreen kit, Thermo fisher) of the islets recovered after perifusion.

**Challenge of islets with inflammatory cytokines**. Human islets (500 IEQ) in 24-well plates 2 ml of PIM(R) media were transfected with the aptamer chimeras 24 h after isolation. Forty-eight hours later, each well was split, and half of the wells were treated with TNF-α, IL-1β, and IFN-γ (1000 U/ml each) (Peprotech, Rocky Hill, NJ), and half were left untreated. Twenty-four hours after the challenge, islets were dissociated and analyzed by flow cytometry.

**Flow cytometry**. Islet cultures were spun down at 250 × g for 6 min. The supernatant was decanted, and islets incubated with 400 µl of 4 °C trypsin for 5–10 min′. The reaction was quenched with 20% FBS containing RPMI. Islets were passed five times through a 5/8 26G needle, and cells spun at 500 × g for 6.5 min and washed with PBS. Cells were stained with the fixable Live Dead dye (Live/dead yellow, ThermoFisher) for 20 min at RT, washed, and incubated for 30 min at 4 °C with cy3-labeled RNA aptamers (38 pmoles/250 IEQ). The cell suspension was then washed once in PBS, spun at 550 × g for 6.5 min, permeabilized and fixed with the 1

Perm/fix solution (BD Bioscience) for 20 min at 4 °C, washed twice with 1× perm/wash buffer, and stained with anti-insulin and anti-glucagon antibodies for 30 min at 4 °C. MIN6 cells were detached with trypsin and incubated for 30′ at 4 C with either aptamer-chimera hybridized to Cy5 guide RNA or biotinylated aptamer complexed with streptavidin-AF647 in HBBS. Samples were washed once again with perm/wash buffer and once with PBS, resuspended in 300 µl of PBS, and analyzed on an LSR2 flow cytometer equipped with 405, 488, 532, and 635 nm lasers (BD Bioscience) or the Cytoflex 18. Data were acquired with FACS DIVA v 8.0.1 or Cytexpert v2.3 software and analyzed using FCS v6 and v7 express plus software (Denovo-Software).

**Aptamer-streptavidin conjugation**. Aptamers biotinylated at the 3′end were synthesized by oligofactory and added (at a 4 to 1 molar ratio) to the streptavidin conjugated with AlexaFluor-647 (Biolegend) or with AlexaFluor-750 (Thermo fisher) gradually (1/6 of the volume added every 5 min). The mixture was incubated ON at RT in rotation in the dark, concentrated with an Amicon Ultra-4 centrifugation filter (50 kDa, Millipore), washed twice with 1 ml PBS, and brought to the 62.5 µM. The aptamer-streptavidin complexes were heated to 65 °C for 10′ and then allowed to cool to room temperature for at least 10 min before. Effective conjugation was evaluated by EMSA on a 2% agarose gel.

**Selection of saRNAxiap**. Bioinformatics screening of the *XIAP* promoter was screened with the algorithm developed by Wang et al.[50]. Briefly, the 1100 bp region between nucleotide 123858524-123859624 of Chromosome X upstream of XIAP transcriptional starting site (−1200 to −100 bp) was retrieved from the ensemble GCA_000001405.28 from the Ensembl Genome Browser and fed to the excel macro developed by Wang et al.[50]. The candidate sequences with Wang's score ≥4 were screened with Blat[96] using default parameters and Blast (with filtered E value <0.5 and >90% identity) to remove those sequences with more than one hit in the genome. This resulted in seventy-five 19-nucleotide long sequences that were synthesized (Sigma-Aldrich) as double-stranded saRNA. All sequences were synthesized with 3′dTdT overhang for empirical validation on A594 cells. In particular, 10⁴ A549 cells cultured O/N at 37 °C in complete media in flat bottom 96 well plate (10⁴ cells/well) and transfected with saRNA (100 nM) using lipofectamine 3000 (Thermo Fisher) following the manufacturer's instruction. Media was changed 30 min and 2 days after transfection. Ninety-six hours after transfection, RNA was isolated with TRIzol, and XIAP expression was quantified by qRT-PCR (Applied Biosystems) and normalized to 18 S expression. The effect size was calculated from triplicates with the following formula: effect size = $(2^{-\Delta CT exp} - 2^{-\Delta CT ctrl})/SD_{exp}$.

**Aptamer-si/saRNA chimera transfection and fluorescent probes**. Aptamer chimeras were generated as previously described[28]. Briefly, aptamer-passenger strand conjugates were produced by PCR and T7RNA polymerase using the Sul5′ and the Sul3′ primers elongated in 5′ with the relevant passenger sequence (Supplementary Table 3). The guide strand RNA sequences were then annealed at an equimolar ratio in a thermocycler using the following conditions: 70 °C for 10 min, cooling to 25 °C at 0.1 °C/s. Annealing was confirmed by gel shift electrophoresis on a 3% agarose gel. Non-dissociated islets (250 IEQ) were transfected by adding the relevant or scrambled si/saRNA-aptamer chimera (150picomoles/100 IEQ for siRNA and 200 picomoles/100 IEQ for saRNA) to human islets in PIM (R) media. Clusterin siRNA was previously validated[97], whereas TMED6 siRNA was selected from the first 200 nt of the TMED6 mRNA (NM_144676.4) using the Invivogen siRNA wizard and validated in vitro. When used as an immunofluorescence probe, aptamer-chimera was annealed to the complementary RNA strand labeled in 5′ with Cy5.

**qRT-PCR**. TMED6 and clusterin expression was evaluated by qRT-PCR using 200 ng of human islets total RNA extracted with Trizol. Data were normalized on 18S expression (Applied biosystem) and expressed as follows: expression = $2^{-(CT exp-CT 18S)} \times 1000$).

**Protein arrays**. HuProt 2.0 arrays (ArrayIt) were prepared for binding as recommended by the manufacturer. The protein surface of the array was deactivated with 3 ml of "Chem block" buffer (Arrayit) for 1 h at RT. After deactivation, the array was blocked with blocking buffer (Arrayit) for 1 h and then hybridized for 90 min at 4 °C with the cy3-labeled aptamer (40 picomoles) in 100 µl of blocking buffer supplemented with BSA (2%, Sigma) and yeast RNA (0.1 mg/ml, Ambion). After hybridization, the arrays were washed (5 times × 5 min) in PBS, dried using the Microarray High-Speed Centrifuge (MHC—ArrayIt), and images were acquired with the Genepix 4000B microarray reader and acquired with GenePix Pro Microarray Analysis Software (Molecular Devices).

**Protein array analysis**. The raw intensity signals were extracted from the gpr files using the *PAA* package (v. 1.0)[98] in R 3.4.4; before preprocessing, each microarray was visually inspected for any artifact using the *plotArray* function. Then, the median fluorescence intensities at 635 nm wavelength were inputted as raw intensity foreground signals; specifically, intensity signals of replicated spots were

mean-centered, and background corrected using default parameters, e.g., the *saddle* variant of the *normexpr* method in the *backgroundCorrect* function, and were finally normalized using the quantile method of the *normalizeArrays* function. Not annotated spots, e.g., without both gene Symbol and Refseq annotations (n.1,975), were excluded from following analysis. Statistically significant differences in aptamer binding to the proteins have been defined on the logged (log2) binding values using the *t* test method in the *diffAnalysis* function and considered significant if the *p* value ≤0.05 and the absolute fold change ≥2. Raw data are available in GEO as GSE162273.

**Gene expression analysis**. Gene expression data was recovered from the GEO datasets GSE2109 and GSE15543, both for the whole pancreas tissues (GSM53046, GSM325790, GSM325838, GSM277701, GSM277726, GSM277736, GSM231922, GSM203675, GSM203703, GSM203761, GSM179781, GSM179869, GSM152744, GSM137958, GSM117645, GSM117647, GSM89045) and for the islets cells (GSM388749, GSM388750, GSM388753, GSM388754, GSM388759, GSM388760, GSM388766, GSM388767) using the *GEOquery* package (v. 2.46.15)[99] in R (v 3.4.4). The raw intensity signals were extracted from CEL files and normalized using the *justRMA* function of the *affy* package (v. 1.56)[100]. Fluorescence intensities were background-adjusted and normalized using the quantile normalization; afterward, log2 expression values were calculated using median polish summarization and custom chip definition files for Human U133 plus2 array based on Entrez genes (HGU133Plus2_Hs_ENTREZG version 20.0.0; 19,363 unique genes) from the Brain Array webpage[101]. Statistically significant differences in gene expression were determined using the moderated *t* test in the *limma* package (v. 3.34.9)[102]; a gene was defined significant if with absolute fold change ≥two and adjusted *p* value ≤0.05 after multiple testing corrections with the Benjamini and Hochberg method, e.g., FDR.

**Aptamer-mediated immunoprecipitation and mass spectrometry**. Human islets (1000 IEQ) were dissociated for 5 min with 0.025% Trypsin EDTA and washed three times with PBS-1%BSA (2 ml, 514 g 6 min). The cell suspensions were then incubated with biotinylated aptamers (100 pmoles) in binding buffer (5 mM $Mg^{2+}$ PBS with masking oligo 1–3 at 5 μM each—Supplementary Table 3) for 30 min at 4 °C. Cells were washed three times with PBS-BSA1%. To separate membranes from intracellular components, cells were incubated in a mild hypotonic lysis buffer containing 10 mM Tris–HCl, 50 mM NaCl, 500 μM MgCl, 1 μM DTT, protease inhibitor cocktail, masking oligo (5 μM), and yeast RNA (4 mg/ml) for 2 min on ice. Immediately after incubation, cells were gently homogenized in a Dounce homogenizer, ten times on ice, mixed with streptavidin-conjugated magnetic beads (Dynabeads, Thermo Fisher) in binding buffer for 1 h at RT in rotation. Beads were magnetically recovered, washed with 1 ml of PBS, resuspended in 30 μl of Laemmli buffer, and captured proteins run on a 4–20% gradient SDS-PAGE. Gels were stained using Brilliant Coomassie Blue (Pierce). Bands from the m12-3773 immune precipitate and corresponding area (i.e., same weight) from the scrambled controls were cut out and sequenced by microcapillary LS/MS/M and analyzed by Mascot software at the University of Kentucky Mass Spectrometry Facility (Supplementary Table 5).

**Mice**. All animal experiments were approved by the Division of Veterinary Resources and the Institutional Animal Care & Use Committee of the University of Miami. Eight to ten weeks old male NOD.Cg-Prkdcscid Il2rgtm1Sug/JicTac (NOG) (Taconic), female Balb/C, and C57Bl/6J mice (Jackson) were purchased, allowed free access to food and water, and were maintained on a 12-h light/dark cycle at room temperature (range 20–23 °C) and controlled humidity (range 30–70%) in individually ventilated cages at the pathogen-free animal facilities at the University of Miami on a chlorophyll free diet. Mice were allowed to acclimate for at least 1 week before starting any experiments. NOG mice were transplanted with variable numbers (150–500 IEQ) of human islets in the EFP[103] or under the kidney capsule. Transplants were performed by the small animal core at the Diabetes Research Institute. Unless otherwise stated, mice were treated with the aptamer at least 21 days after transplant. BALB/c mice were transplanted with B6 and Balb/c islets (250 IEQ each) dorsally in the flank using a thrombin-plasma biological scaffold as described[104].

**IVIS analysis**. Isofluorane-anesthetized, islet-transplanted mice were analyzed by the In Vivo Imaging System (Xenogen IVIS Spectrum Perkin Elmer) 4 h after i.v. injection of 12.5 pmol/g of aptamers conjugated with either streptavidin-Alexa Fluor-750 or streptavidin-Alexa Fluor-647. Fluorescence was quantified with the Living Image v4.3 software (Perkin Elmer). The signal to background ratio was calculated by evaluating the radiance in the region of interest (i.e., EFP or liver area) over an ROI of the same size drawn on the lungs region after subtracting the autofluorescence signal from non-injected mice.

**Marginal mass islet transplantation experiments**. Diabetes was induced in 10–16 weeks old immunodeficient NOG female mice (Taconic, Rensselaer, NY) with 5 low doses of streptozotocin (50 μg/g, i.p., q.d.). Mice were kept euglycemic with a subcutaneous insulin pellet until human islets were available (~5–20 days). Forty-eight hours before the scheduled transplantation, the insulin pellet was

removed. Blood glucose was monitored, and only hyperglycemic mice were used for the experiments. On the day of the transplantation, mice were anesthetized and transplanted by the DRI small animal core under the kidney capsule as previously described[105] with 500 IEQ of human islets treated with aptamer chimeras or left untreated. To evaluate the quality of the preparation, 1–2 mice were transplanted with 1200 IEQ from each preparation as positive controls. Blood glucose was monitored by venipuncture three times a week.

**Statistical analysis**. Sigmaplot 12.5 (Systat Software) was used for data analysis. Statistical tests (one-way ANOVA followed by Holm–Sidak test for multiple pairwise comparisons or student *T* test) were applied as indicated in the figure legends in a two-sided, unpaired fashion after normality was evaluated by the Shapiro–Wilk test. In vitro analyses and in vivo experiments were repeated two to five times to ensure reproducible conclusions; the exact number of repetitions is stated in each figure legend. Log-rank test was used for survival analysis followed by all pairwise multiple comparison procedures (Holm–Sidak method). Data from multiple experiments were cumulated unless otherwise indicated in the figure legends. No experimental data point was excluded from the analyses. The sample size was chosen by power analysis using effect size determined by pilot experiments.

**Reporting summary**. Further information on experimental design is available in the Nature Research Reporting Summary linked to this paper.

## Data availability
Protein chip data are available in GEO as GSE162273.

HT-sequencing data from SELEX experiments are available in GEO as GSE197262

The following series and datasets from GEO have been used in this study: series GSE2109 datasets # GSM53046, GSM325790, GSM325838, GSM277701, GSM277726, GSM277736, GSM231922, GSM203675, GSM203703, GSM203761, GSM179781, GSM179869, GSM152744, GSM137958, GSM117645, GSM117647, GSM89045, and series GSE15543[106] datasets # GSM388749, GSM388750, GSM388753, GSM388754, GSM388759, GSM388760, GSM388766, GSM388767.

Data from the experiments are provided in the Supplementary Information. Source data are provided with this paper.

## Code availability
No new codes were used in this study.

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

## Acknowledgements

The authors thank the small animal core, the flow cytometry core, and the GMP at the Diabetes Research Institute for the help with mouse surgery, islet cell analysis, and human islet isolation, respectively; Dr. Szust for the help in islets transplantation experiments and Drs. Malek and Pugliese for the critical reading of the paper and the helpful discussions. This research was performed using funds from the JDRF awards 17-2013-326 and SRA-2020-962-S-B to P.S., the JDRF award 2016-173-A-N to D.V.S., from Fondazione AIRC under 5 per Mille 2019 program (ID. 22759) to S.B., and the NIDDK-supported Human Islet Research Network (HIRN, RRID:SCR_014393; https://hirnetwork.org; UC4 DK 116241 to P.S.).

## Author contributions

Conceptualization: P.S.; Methodology: P.S.; Software: J.C., A.G., and S.B.; Investigation: A.D.F., A.Z., D.V.S., S.Z., J.C., O.A., V.K. and P.S.; Validation: A.D.F., A.Z. D.V.S., and P.S.; Formal analysis: P.S. P.B.; Visualization: P.S., D.V.S., S.Z. and A.Z.; Writing the first draft: P.S., Supervision: P.S.; Project administration: P.S.; Funding Acquisition: D.V.S., P.S. All authors contributed to the writing and read and approved the final paper.

## Competing interests

Some of the authors of this paper (A.D.F., A.Z., D.V.S., P.S., S.B., J.C., A.G., and S.Z.) are named inventors for a patent filed by the University of Miami on the use of RNA aptamers for the detection and targeting of human islets. The remaining authors declare no competing interests.
