## [Peer Review File · Nature Communications]

Reviewers' comments:

Reviewer #1 (Remarks to the Author):

In the present study Vam Simaey and colleagues identified two aptamers specific to human and mouse beta cells using for the first time two SELEX methods on cluster cells (in this case, pancreatic islets). The target proteins for the candidate aptamers were identified as Transmembrane p24 trafficking protein 6 and Clusterin. These probes are reasonably specific to the beta cells and the authors used them to successfully image human and/or mouse islets grafted in immune deficient mice and to deliver an anti-apoptotic protein (XIAP). The data are potentially interesting and of relevance for the beta cell imaging and therapeutic fields. However, additional evidence must be provided to support the Author's claims and to increase the impact of study.

Specific points:

Figure 1b: According to the Material and Methods, aptamer libraries are tested through multiples cycles of selection. As a consequence, the library complexity decreases after each cycle. An increase in library complexity, however, was observed after cycle 6. An explanation should be provided for this phenomenon.

HT-cell SELEX approach: This method failed to identify a specific aptamer and was essentially used to confirm the identity of aptamers identified using HT-toggle SELEX. Taking this into account, the corresponding description could be shortened.

Figure 2a: Clear (but limited) exocrine signal is also visible for m12-3773 aptamer. Based on figures 2a and 2b, only aptamer 1-717 seems sufficiently specific to the islets. Of note, different names are used for the same aptamers and could lead to confusion (m12-3773, m12-2773 in ESM).

Aptamer affinity: The KD value for TMED6 aptamer is close to 109 M which could indeed be sufficient for future in vivo imaging studies. Of note, the affinity of aptamers for clusterin should also be provided.

Figure 2c: A clear signal can be observed in the exocrine tissue for both aptamers. This should be more clearly acknowledged as it may compromise the chances to accurately quantify beta cells by in vivo imaging.

The use of a high content imager is interesting, but the analysis could be complicated and depends on the quality of the segmentation. It would be important to show some segmentation examples as Supplementary Material. The Authors also mentioned on page 26 that the pictures have been optimized using ImageJ. Could the authors give more details about the procedure and examples of pictures before/after the treatment? Modification of pictures should be clearly mentioned, explained and justified.

Figure 2d-e: Based on flow cytometry and confocal microscopy dot plots, it seems that aptamers stains both alpha and beta cells. Quantification of the percentages of alpha/beta cells should be provided. The

right panel also shows a more important signal in beta cells than in alpha cells and does not correlate well with the histograms. The background signal given by alpha and beta are also very different. Are these data normalized for the background signal? Importantly, the beta/alpha cell ratio seems also much lower than expected for human islets. Human islets are thought to be composed of at least 50% of beta cells but authors only identify around 20% of cells as beta cells.

Figure 3a: The authors should confirm that the isoforms targeted by the aptamers are indeed beta cell- or islet-specific, ideally by RT-qPCR.

Figure 3b: What was the normalization method used for RT-qPCR? Use of arbitrary units is not a usual procedure. What was the transfection efficiency reached using aptamers?

Figure 4c and Figure 5e: The authors assume that the aptamer uptake in the upper abdomen area in Figure 4c is associated to the binding of aptamers to endogenous islets. However, such uptake is not visible in Figure 5e. Fluorescence imaging was also performed at different time points: 4h and 6h.

- It would be important to demonstrate that the uptake in the pancreas-surrounding area is really pancreas specific as the findings observed could also be explained by liver retention of the aptamers. Determining ex vivo biodistribution data could be useful. Authors could use radiolabeled aptamers if fluorescence is an obstacle.
- The pharmacokinetic profiles of the selected aptamers and how the imaging time windows have been selected should be shown. For how long do they remain in circulation? How are they excreted?
- Details regarding the quantification should be provided (formula, normalization method, organ segmentation if any, etc.).
- Additional controls could be useful: signal in non-transplanted mice, signal in mice transplanted and injected with a non-relevant control aptamer etc.

Figure 5a: The aptamers seem to be more specific to mouse than to human islets (Figures 5a-d vs Figure 2c). Exocrine staining is mentioned by the Authors, but the pictures presented show very little uptake in this tissue compared to Figure 2c.

Figure 5c-d: It is unclear why do the authors quantify the uptake of the aptamers in mouse pancreas on dissected tissue. According to figure 4c and the Author's statements, endogenous islets should be detectable by fluorescence imaging.

Figure 5e: Grafting of allogenic tissue triggers a rapid inflammatory response, which could affect the functionality and the vasculature of the graft, which in turn could affect aptamer uptake. The signal in the syngeneic graft also seems to increase according to the time (due to an increase in graft vascularization? An increase in receptor expression?). This should be commented upon.

As described in the text, normalization was performed using non-transplanted animals. A normalization procedure must take into consideration the quantity of aptamer injected in each mouse, as well as the non-specific signal in a selected non-targeted tissue.

Figure 5f: The inter-experiment variability seems important. A third experiment could be useful and

would allow performance of statistical analysis.

Figure 6d:

- The efficiency of the transfection in the samples should be provided.
- The expression of genes related to apoptosis in beta cells (e.g. Bcl2, BclXL, Bim etc) should be quantified.

Additional points:

- It would be important to perform in vivo competition studies (using for instance excess of non-labelled aptamers).
- The toxicity of the aptamers identified by the authors should be excluded on mouse and human islets. Cell viability and insulin secretion capacity should be assessed after a 24-28h culture in presence of the aptamers at concentrations at least 10-times higher than those observed in vivo.

Minor points

- The Authors should perform a detailed revision of the text, as there are several mistakes (see examples below).
- In the Supplemental figures, it should be "Adrenal" and not "Adreanal";
- The glucose concentration of the PIM(R) media used for the culture of human and mouse islets should be indicated;
- In Methods, "Aptamer streptadavin conjugation", write "...AlexaFluor-647 (Biolegend) AND with Alexa-Fluor-750..."
- In Methods, "Mice", indicate "...were purchased AND maintainED in the pathogen free..."
- In the Legend to Figure 1, use "enrichment" and not "enrichement"; use "aptamerS", instead of "aptamer" after c).
- In Figure 2, write "glucagon" and not "glucagone".
- In Figure 3, f) "...was admixed (and not admixes)..."

Reviewer #2 (Remarks to the Author):

The study by Simaey et al. demonstrates the selection of 2'fRNA aptamers targeting islet cells. They employed two selection strategies to generate aptamer that recognise mice and human islet cells, underwent extensive NGS analysis and screening of monoclonal sequences that fulfil certain interaction criteria. These experiments ultimately lead to the identification of two aptamers, m12-3773 and aptamer 1-717 which were further studied. Essentially, the putative target molecules of the two aptamers were identified by MS analysis or protein microarrays. siRNA conjugates of the aptamer were then employed for specific knockdown of siRNA targeted genes. Although the overall study seems to be sound and interesting, the experiments regarding the aptamers in the ms need major improvements:

- Most importantly, all experiments in which the two aptamers are used lack control experiments with scrambled, non-binding variants of the two aptamers. These controls are mandatory, especially for siRNA delivery, protein arrays and MS analysis and all in vivo studies. Non-binding controls were only

used when comparing enriched vs non-enriched libraries. But in all subsequent experiments these controls are missing. Scrambled variants have the same nucleotide composition as the aptamer but a different nucleotide distribution.

- MS for target ID: This section lacks experimental results in terms of gel pictures, of controls done, and a list of targets pulled from the samples is not shown (I doubt that only the putative indicated target was identified, but also other proteins might be co-enriched by the pull-down experiments)? How was the actual target chosen if there were more than one protein in the pulled-down soup? How does blocking of the 3'-end (constant region) of the aptamers with a complementary oligo interfere with target binding on tissue level? These questions are not addressed in the ms but are important to validate the study.

- Same is true for the protein arrays applied to target ID. The actual primary data are neither shown in the ms nor in the supporting part. Why is that? The reader might be interested in the specific pattern of the aptamers on the array compared to non-binding controls and other positive controls, e.g. including known aptamer-target pairs to validate the system. This is important as the approach taken by the authors seems to be not very common in the field, but interesting.

Reviewer #3 (Remarks to the Author):

The manuscript by Dimitri Van Simaey et al report the identification and characterization of two RNA aptamers, with putative specificity for TMED6 and clusterin, that recognize both human and mouse β cells with high selectivity in vitro and in vivo. These aptamers facilitate the quantitative measurement of human β cell mass not only after engraftment in immune deficient mice but also during the rejection of islet allografts in immune competent mice. These aptamers allow also the efficient and non-viral delivery of saRNA able to upregulate the antiapoptotic gene xiap in human nondissociated islet.

Overall, the study seems to have been well conducted. The methods and results are clearly described. The study is very relevant in the field and can contribute a lot for the future research.

Two major points should be better discussed:

- TMED6 was previously described highly expressed in exocrine tissue (see protein atlas <https://www.proteinatlas.org/ENSG00000157315-TMED6/tissue/pancreas>). How much relevant is this for the possibility to use this reagent in humans?

- clusterin is an extracellular protein secreted by various tissues and was detected in blood plasma, cerebrospinal fluid, milk, seminal plasma and colon mucosa. It functions as extracellular chaperone that prevents aggregation of non native proteins. It was detected in brain, testis, ovary, liver and pancreas, and at lower levels in kidney, heart, spleen and lung. How much relevant is this for the possibility to use this reagent in humans?

Reviewer #1 (Remarks to the Author):

In the present study Vam Simaey and colleagues identified two aptamers specific to human and mouse beta cells using for the first time two SELEX methods on cluster cells (in this case, pancreatic islets). The target proteins for the candidate aptamers were identified as Transmembrane p24 trafficking protein 6 and Clusterin. These probes are reasonably specific to the beta cells and the authors used them to successfully image human and/or mouse islets grafted in immune deficient mice and to deliver an anti-apoptotic protein (XIAP). The data are potentially interesting and of relevance for the beta cell imaging and therapeutic fields.

Thank you for the positive and constructive comments. We agree that the article is of interest for the broad readership of *Nature Communication*.

Specific points:

1.1 Figure 1b: According to the Material and Methods, aptamer libraries are tested through multiples cycles of selection. As a consequence, the library complexity decreases after each cycle. An increase in library complexity, however, was observed after cycle 6. An explanation should be provided for this phenomenon.

We agree with the reviewer that the increase in complexity after cycle 6 is unusual. *We revised the sequencing data and the graph reflect the actual data.* Although many speculative interpretations might be given, the decrease in complexity on cycle 6 and its subsequent increase in cycle 7 and 8, this might be related to some peculiar characteristic of the 3rd islets donors that 2 days after isolation might not have expressed all the epitope expressed in the other islets. This may have caused a decrease in frequency of some clusters and consequently library complexity. With the islets from a new donor (cycle 7 and 8) these epitopes might be expressed, and complexity seemed to stabilize. Unfortunately, we do not have any tissues left from the donors thus we cannot test any hypothesis.

1.2 HT-cell SELEX approach: This method failed to identify a specific aptamer and was essentially used to confirm the identity of aptamers identified using HT-toggle SELEX. Taking this into account, the corresponding description could be shortened.

We respectfully disagree. Although it is true that the polyclonal library from the HT-Cell SELEX failed to provide sufficient signaling in immune fluorescence to identify human islets on tissue, 4 of the 9 monoclonal aptamers (including aptamer 1-717 evaluated in the study) derived from HT-Cell SELEX did recognize human islets (Fig. 2). Additionally, many other monoclonal aptamers identified with HT-SELEX clustered with the ones identified by the Toggle SELEX suggesting a convergent selection toward similar aptamers. Thus, although the polyclonal library from HT-SELEX was unable to give sufficient signal in the islets to be detected, many aptamers specific for islets cells were significantly expanded and identifiable by HT-Sequencing and subsequent empirical screening.

1.3 Figure 2a: Clear (but limited) exocrine signal is also visible for m12-3773 aptamer. Based on figures 2a and 2b, only aptamer 1-717 seems sufficiently specific to the islets.

We agree that a few cells in the exocrine pancreas of the example provided are positive for aptamer m12-3773. We realized that the close-up pictures we provided were not giving sufficient information on the major finding. While a few acinar cells close to the islets are positive, the large majority of the acinar cells (>95%) are not recognized by aptamer m12-3773. To better represent the specificity, in the revised version, we provide as Supplementary Fig. 3 the scan of a human pancreatic section stained with the aptamers and antibodies against insulin and glucagon. You may notice that aptamer m12-3773 has a good specificity for the islets although some positive binding is appreciable in cells near the islets. This can be related to the possibility that clusterin (the putative target of this aptamer) can be secreted and taken up by nearby cells. Image cytometry analysis of these data is now also provided in the revised Fig.3c by gating on islet or on acinar tissue as detailed in the new Supplementary Fig. 4. We also realized that the positioning of the color grid plot made following the tissue specificity difficult. We moved aptamer 1-717 and m12-3773 to the right to make the figure interpretation easier to follow.

1.4 Of note, different names are used for the same aptamers and could lead to confusion (m12-3773, m12-2773 in ESM).

Thank you for noticing that. Those were typographic errors that have been corrected. The right clone name is m12-3773.

1.5 Aptamer affinity: The KD value for TMED6 aptamer is close to 109 M which could indeed be sufficient for future in vivo imaging studies. Of note, the affinity of aptamers for clusterin should also be provided.

Unfortunately, we found that aptamer binding to recombinant clusterin is low and dependent on production lot and manufacturer. This might be due to the posttranslational modification of clusterin in the islet making it specific but also making its KD determination extremely difficult. To overcome these difficulties, we determine the KD by staining single cell suspensions of human islets with different concentration of each aptamer. These new data have been included in Supplementary Fig. 5 and have been mentioned in the main text on page 7. Of note, the flow

cytometry approach shows a lower affinity than the SPR approach possibly because of the random conjugation of the Cy3 fluorochrome that in some position may interfere with aptamer binding. This has been noted on page 8 of the revised manuscript.

1.6 Figure 2c: A clear signal can be observed in the exocrine tissue for both aptamers. This should be more clearly acknowledged as it may compromise the chances to accurately quantify beta cells by *in vivo* imaging.

This has been acknowledged more clearly on page 7.

1.7 The use of a high content imager is interesting, but the analysis could be complicated and depends on the quality of the segmentation. I would be important to show some segmentation examples as Supplementary Material.

We added examples of segmentation and image cytometry workflow in Supplementary Fig. 4.

1.8 The Authors also mentioned on page 26 that the pictures have been optimized using ImageJ. Could the authors give more details about the procedure and examples of pictures before/after the treatment? Modification of pictures should be clearly mentioned, explained and justified.

We apologize for the lack of clarity in the previous version. Images were processed in ImageJ with CLAHE to remove the artifactual different illumination and staining especially in the DAPI channel that can hinder a correct segmentation of scanned images. This has been clearly mentioned and described in the material and methods (page 28 of the revised manuscript).

1.9 Figure 2d-e: Based on flow cytometry and confocal microscopy dot plots, it seems that aptamers stains both alpha and beta cells. Quantification of the percentages of alpha/beta cells should be provided. The right panel also shows a more important signal in beta cells than in alpha cells and does not correlate well with the histograms. The background signal given by alpha and beta are also very different. Are these data normalized for the background signal?

Thank you for noticing the omission in the data transformation. All the flow cytometry experiments were performed with the scrambled aptamers and vital dye because of the higher propensity of these cells to die because of isolation and islets dissociation. Data have been normalized by subtracting the non-specific signal from the scrambled staining. This has been specified in title axes of the spaghetti plot.

Importantly, the beta/alpha cell ratio seems also much lower than expected for human islets. Human islets are thought to be composed of at least 50% of beta cells but authors only identify around 20% of cells as beta cells.

While in image cytometry performed on slides, we do find a 2/1 - 3/1 beta/alpha ratio, in flow cytometry, we normally find a 1/1 - 1/2 beta/alpha ratio especially when only viable (vital dye negative cells) are considered. This beta cell loss is possibly due to the isolation and dissociation procedures as well as the presence of a higher number of beta cells in the duplets. Please note that since aptamer stainings are performed on the same preparation, this beta cell loss does not affect the findings.

2.0 Figure 3a: The authors should confirm that the isoforms targeted by the aptamers are indeed beta cell- or islet-specific, ideally by RT-qPCR.

Tissue distribution of islet specific TMED6 is confirmed by a previous paper (Wang et al Pancreas. 2012 Jan;41(1):10-4.). Our transcriptome analysis using public database (figure for reviewers #1) or comparing the transcriptome of whole pancreas (Supplementary Table 6) with islets confirm this finding.

Unfortunately, the study of clusterin specific proteoforms is much more complex: although 3 RNA isoforms are present, most of the variability of clusterin arises from post-translational modifications. Indeed, we observed a strong variation of aptamer m12-3773 binding to different lots of recombinant clusterin from HEK293 cells and no binding to clusterin produced in bacteria, suggesting that our aptamers recognize a post-translationally modified epitope. Because of the complexity of the isolation and characterization of the relevant epitopes and its analysis across human tissues, we believe that this should be focused on in further studies, as it requires multiple experiments and figures. This has been mentioned in the discussion.

2.1 Figure 3b: What was the normalization method used for RT-qPCR? Use of arbitrary units is not a usual procedure. What was the transfection efficiency reached using aptamers?

Data from qRT-PCR have been expressed as $2^{-(CT_{exp}-CT_{18S})} \times 1000$. This has been specified now in the Material and methods section (page 31).

2.2 Figure 4c and Figure 5e: The authors assume that the aptamer uptake in the upper abdomen area in Figure 4c is associated to the binding of aptamers to endogenous islets.

We apologize for not being sufficiently clear in the previous version of the paper. IVIS does not have the resolution to discriminate the passive aptamer trapping in the liver from the one in the pancreas. For this reason, human islets were engrafted in the epididymal fat pad, and that region was considered as ROI in Fig. 4c. Indeed, signal in the upper region is visible when aptamer is injected in mice that did not received islet graft (0 IEQ) (Fig. 4c of the previous version and Fig. 4d of the current version) and in mice injected with scrambled aptamer (Fig. 4d). This has been made clearer in the text and by adding the ROI regions with the legend in the figure.

2.3 However, such uptake is not visible in Figure 5e. Fluorescence imaging was also performed at different time points: 4h and 6h.

In Fig. 5e, mouse islets were engrafted subcutaneously dorsally. Images were taken dorsally and that is sufficient to mask the abdominal signal from the liver.

2.4- It would be important to demonstrate that the uptake in the pancreas-surrounding area is really pancreas specific as the findings observed could also be explained by liver retention of the aptamers. Determining ex vivo biodistribution data could be useful. Authors could use radiolabeled aptamers if fluorescence is an obstacle.

Since we have recently shown that fluorochrome and streptavidin are the major factors responsible for liver entrapping (De La Fuente et al. Sci. Transl. Med. 2020, 12(548), eaav9760), we have performed an aptamer biodistribution using unconjugated aptamers 24 hours after

injection and qRT-PCR for aptamer detection. Briefly, human islets were engrafted under the kidney capsule, and 21 days later, a mixture of the two aptamers was given i.v. Aptamers were quantified by qRT-PCR using known aptamers quantities as standard curves. Please see also answer in 2.2.

2.5- The pharmacokinetic profiles of the selected aptamers and how the imaging time windows have been selected should be shown. For how long do they remain in circulation? How are they excreted?

We agree that pharmacokinetic studies using radiolabeled aptamers is of interest, but we believe that it deserves more experiments using aptamers and radio-labelling suitable for clinical use, since formulation does impact both biodistribution and PK. Indeed, we are adapting the aptamer for the clinical use in association with PET. This however is being delayed by COVID-related restrictions. Thus, these studies will be performed during the preclinical development of these reagents and will be material for follow up publications.

2.6- Details regarding the quantification should be provided (formula, normalization method, organ segmentation if any, etc.).

Details have been now provided in the Material and methods section (page 34), by adding the following paragraph: “*Signal to background ratio was calculated by evaluating the radiance in the region of interest (i.e., EFP or liver area) over an ROI of the same size drawn on the lungs region after subtracting the autofluorescence signal from non-injected mice*”

2.7- Additional controls could be useful: signal in non-transplanted mice, signal in mice transplanted and injected with a non-relevant control aptamer etc.

In the revised version of the paper, we included: i) Images of mice injected with scrambled aptamers in Fig. 4d, ii) the IVIS results from different mice injected with scrambled aptamers in Fig. 4b, and iii) data from the qRT-PCR based biodistribution analysis (Fig. 4A). Animals that underwent sham surgery were already provided and labeled as “0 IEQ”. Overall the signal from the scrambled aptamers in the graft region is negligible while important signal is noticeable in the liver area.

2.8 Figure 5a: The aptamers seem to be more specific to mouse than to human islets (Figures 5a-d vs Figure 2c). Exocrine staining is mentioned by the Authors, but the pictures presented show very little uptake in this tissue compared to Figure 2c.

As mentioned above, it is difficult to compare the specificity evaluating a close-up of a single islet. We realized that this has been misleading. The additional data provided in this version clearly indicate that both aptamers are quite specific for both mouse and human islets compared to the acinar tissues. It is important to note that across tissues, our aptamers appear to be more specific for human islets than mouse islets.

2.9 Figure 5c-d: It is unclear why do the authors quantify the uptake of the aptamers in mouse pancreas on dissected tissue. According to figure 4c and the Author's statements, endogenous

islets should be detectable by fluorescence imaging.

Please see response 2.2 above. We cannot detect endogenous islets by IVIS because of the low resolution of the technique. This is why we are moving to PET, and the very reason why pancreas was evaluated on dissected tissues and by in vivo labeling with AF647 and ex vivo counterstaining with anti-insulin and anti-glucagon antibodies.

3.0 Figure 5e: Grafting of allogenic tissue triggers a rapid inflammatory response, which could affect the functionality and the vasculature of the graft, which in turn could affect aptamer uptake. The signal in the syngeneic graft also seems to increase according to the time (due to an increase in graft vascularization? An increase in receptor expression?). This should be commented upon. We agree that it seems that there might be an increase; however, since it does not reach statistical significance, we do not feel that it should be highlighted.

3.1 As described in the text, normalization was performed using non-transplanted animals. A normalization procedure must take into consideration the quantity of aptamer injected in each mouse, as well as the non-specific signal in a selected non-targeted tissue.

We realized that we were not sufficiently clear, and details were missing in the normalization procedures. This has been rectified now, see 2.6.

3.2 Figure 5f: The inter-experiment variability seems important. A third experiment could be useful and would allow performance of statistical analysis.

Unfortunately, we had to limit the experiments that could be performed for this reply because of the associated cost and the limits imposed by the COVID pandemic. We agree with the reviewer that in one experiment the variation between mice was higher making the rejection less clear. Statistical analysis of the rejection has been already performed (Fig. 5g).

3.3 Figure 6d:- The efficiency of the transfection in the samples should be provided.

In vitro transfection efficiency of beta cells in undissociated islets using Cy3-labeled siRNAs or saRNA is higher than 80% or 90% see figure for reviewer 2a. Functional efficiency depends on the genes and therapeutic RNA, and can vary between 60 and 80%. For example we can silence insulin of more than 90% in non-dissociated mouse islets. (see Fig. 2b for reviewers). We believe that adding this piece of information to the manuscript is redundant because similar functional efficiency is shown in Fig. 3b and h when we silence TMED6 or clusterin in non-dissociated human islets

3.4- The expression of genes related to apoptosis in beta cells (e.g. Bcl2, BclXI, Bim etc) should be quantified.

We respectfully disagree since (1) the pathways that XIAP inhibit have been extensively studied and (2) because functional in vitro and in vivo experiments, such as the one in Fig. 6d-e using islets from nine cadaveric donors, are more indicative of aptamer chimera activity than the measurement of surrogates of apoptosis.

3.5 Additional points:- It would be important to perform in vivo competition studies (using for instance excess of non-labelled aptamers).

Competition studies using recombinant TMED have already been performed in vitro. Doing that in vivo would require massive amounts of proteins at excessive costs. We feel that the use of excess unlabeled aptamer (10-100 time) would provide only little information on aptamer specificity that do not justify the costs of the experiment.

3.6- The toxicity of the aptamers identified by the authors should be excluded on mouse and human islets. Cell viability and insulin secretion capacity should be assessed after a 24-28h culture in presence of the aptamers at concentrations at least 10-times higher than those observed in vivo.

Thank you for the suggestion. We agree that evaluating islets viability and function after incubation with the aptamers is important especially because TMED6 and clusterin have been implicated with insulin secretion and beta cell survival, respectively. The experiment was performed as suggested by incubating human islets with a dose 10 times higher than the one used in vivo and by performing viability and dynamic insulin secretion experiments. These data are included in Supplementary Fig. 7b,c and show that aptamers do not affect islet viability and insulin secretion. Additionally, we evaluate signs of in vivo toxicity in mice injected 5 days with double dose of RNA aptamers (Supplementary Fig. 7a). No differences from control (i.e., mice injected with PBS or with scrambled aptamers were found).

Minor points- The Authors should perform a detailed revision of the text, as there are several mistakes.

Text has been revised and a number of typographical and grammatical errors have been corrected.

Reviewer #2 (Remarks to the Author):

The study by Simaey et al. demonstrates the selection of 2'fRNA aptamers targeting islet cells. They employed two selection strategies to generate aptamer that recognise mice and human islet cells, underwent extensive NGS analysis and screening of monoclonal sequences that fulfil certain interaction criteria. These experiments ultimately lead to the identification of two aptamers, m12-3773 and aptamer 1-717 which were further studied. Essentially, the putative target molecules of the two aptamers were identified by MS analysis or protein microarrays. siRNA conjugates of the aptamer were then employed for specific knockdown of siRNA targeted genes. Although the overall study seems to be sound and interesting, the experiments regarding the aptamers in the ms need major improvements:

Thank you for your comments. The manuscript has been extensively reviewed and we believed improved from the previous version.

1.0- Most importantly, all experiments in which the two aptamers are used lack control experiments with scrambled, non-binding variants of the two aptamers. These controls are mandatory, especially for siRNA delivery, protein arrays and MS analysis and all in vivo studies.

Non-binding controls were only used when comparing enriched vs non-enriched libraries. But in all subsequent experiments these controls are missing. Scrambled variants have the same nucleotide composition as the aptamer but a different nucleotide distribution.

Missing controls have been added in all the figures including staining. Please see also answer to question 2.7 of reviewer 1

1.1 - MS for target ID: This section lacks experimental results in terms of gel pictures, of controls done, and a list of targets pulled from the samples is not shown (I doubt that only the putative indicated target was identified, but also other proteins might be co-enriched by the pull-down experiments)? How was the actual target chosen if there were more than one protein in the pulled-down soup?

We agree that important details and raw data from the I.P./mass spec data were missing. Picture of the gel from one of the experiments from which of the bands were cut has been included in Fig. 3A, and a table including the raw data from the two i.p./mass spec experiments provided in the Supplementary Table 5. Clusterin was the only protein with Mascot score >50 that was pulled down by m12-3773 in both experiments and that was absent in the scrambled aptamers pull downs.

1.1b How does blocking of the 3'-end (constant region) of the aptamers with a complementary oligo interfere with target binding on tissue level? These questions are not addressed in the ms but are important to validate the study.

Generally, the constant region of the aptamer is not involved in the binding (see for example review from Meyer and Levy 2016 (*Mol. Therap. Method. Clin. Develop.* 2016, 5, 16014; doi:10.1038/mtm.2016.14). For aptamer m12-3773, secondary structure analysis indicates that the 3' end does not form any secondary structure and thus should not be involved in binding.

1.1c- Same is true for the protein arrays applied to target ID.

The actual primary data are neither shown in the ms nor in the supporting part. Why is that? The reader might be interest in the specific pattern of the aptamers on the array compared to non-binding controls and other positive controls, e.g. including known aptamer-target pairs to validate the system. This is important as the approach taken by the authors seems to be not very common in the field, but interesting.

An Excel file comprising both protein array and GSE was included as Supplementary Table 6. Raw data have been submitted to geo as GSE162273 and will be made publicly available upon manuscript acceptance. Meanwhile, to revise the raw data please click the following link and enter the token abuxkyugpjezjep into the appropriate box.

<https://www.ncbi.nlm.nih.gov/geo/query/acc.cgi?acc=GSE162273>

Reviewer #3 (Remarks to the Author):

The manuscript by Dimitri Van Simaeys et al report the identification and characterization of two RNA aptamers, with putative specificity for TMED6 and clusterin, that recognize both human and mouse β cells with high selectivity in vitro and in vivo. These aptamers facilitate the quantitative measurement of human β cell mass not only after engraftment in immune deficient mice but also

during the rejection of islet allografts in immune competent mice. These aptamers allow also the efficient and non-viral delivery of saRNA able to upregulate the antiapoptotic gene xiap in human nondissociated islet. Overall, the study seems to have been well conducted. The methods and results are clearly described. The study is very relevant in the field and can contribute a lot for the future research.

Thank you for the positive comments. We hope that the included revisions have further improved the manuscript.

Two major points should be better discussed:

1. TMED6 was previously described highly expressed in exocrine tissue (see protein atlas <https://www.proteinatlas.org/ENSG00000157315-TMED6/tissue/pancreas>). How much relevant is this for the possibility to use this reagent in humans?

Although we do like the efforts of the Protein Atlas, many antibodies are still poorly validated, and HPA012532 is a polyclonal antibody affinity purified and raised using almost the whole protein. The antibody has not been validated in any peer reviewed publication; thus, we do not know which assays have been performed to validate it. Additionally, the pattern of expression of the antibody does not correlate with the one seen evaluating the RNA level. See data for Fig.1 for reviewer. Other non-commercially available antibodies (<https://www.ncbi.nlm.nih.gov/pmc/articles/PMC3870856/>) show good specificity for human islets, and the same studies implicate TMED6 in insulin secretion. Our aptamer has been validated by both silencing experiments and could target inhibition assays. Thus, it has been subjected to validation assays that rarely are performed for antibodies.

2- clusterin is an extracellular protein secreted by various tissues and was detected in blood plasma, cerebrospinal fluid, milk, seminal plasma and colon mucosa. It functions as extracellular chaperone that prevents aggregation of non native proteins. It was detected in brain, testis, ovary, liver and pancreas, and at lower levels in kidney, heart, spleen and lung. How much relevant is this for the possibility to use this reagent in humans?

Additional discussion on clusterin highlighting its extraordinary variability in proteoforms across tissues has been added to the manuscript.

REVIEWER COMMENTS

Reviewer #2 (Remarks to the Author):

The revised manuscript improved compared to the previous version. The authors addressed most of the issues raised by the reviewers. However, the new Fig. 3a showing the gel of the pull down experiment is not very clear. To this reviewer the circled regions have been cut out and analyzed by LC/MS, which is kind of logical. But the gel is not labelled appropriately, e.g. the lane next to the ladder also reveals a band migrating similar to the one shown in the next lane (circled band). What has been loaded in this lane. Likewise, the scrambled control shows a faint band and smear in this area which was also circled cut out. What was identified in this band? How was the gel stained? The methods section does not reveal the staining procedure. The gel also do not reveal the streptavidin monomer bands, which shall occur at 15kDa. The overall quality of the gel is poor, signal noise ratio is limited.

Reviewer #4 (Remarks to the Author):

The authors have responded satisfactorily to most of the issues brought up by Reviewer However, the issues raised regarding probe beta cell specificity and affinity were not addressed sufficiently, and at least the potential impact of these drawbacks should be highlighted in the discussion section.

1. Aptamer beta cell specificity:

Both Apt 1-717 and Apt m12-3773 labels both islets but also some exocrine tissue. As the authors points out in the introduction, novel agents with improved beta cell selectivity is required compared to existing imaging agents. However, the results seen here in eg Fig 1F, Fig 2D, Fig 5A, Suppl Fig 3A-B, Suppl Fig 6 clearly demonstrates labelling of exocrine cells. Additionally, consistently only a subset of insulin positive cells are stained also with the aptamers.

The limitation of the specificity should be discussed more clearly in the manuscript eg how will targeting of some exocrine tissue (and only targeting some beta cells) in affect potential applications.

Any signal from the exocrine pancreas, will most likely mask the in vivo islet specific signal for an imaging probe (as the islets are diluted 50-100 times).

2. Aptamer affinity:

Apt 1-717, which putative target is TMED6, has two different affinities given - in the 0.1 μ M range (Suppl Fig 5, by fluorescent staining) and in the nano molar range (Fig 3G, by SPR).

Here, the SPR fit seems poor (Fig 3G), especially for the dissociation rate (kd) which drives the nanomolar value of the dissociation constant KD. In fact the lowest concentrations are way above the estimated KD. So I would assume the affinity from Suppl Fig 5 is the most reliable, again in the 0.1 μ M range.

A KD in the 0.1 μ M range is definitely not sufficient for in vivo PET imaging, especially when attempting to image a diluted tissue like the pancreatic beta cells.

The same is true for Apt m12-3773.

Thus, the fact that the relatively poor affinity of the probes (with respect to imaging) will make PET

imaging more or less impossible should be addressed in the discussion. E.g. probe improvement is required before further progress in SPECT/PET in vivo imaging. I suggest to show Suppl Fig 5 in the main manuscript and instead move the SPR Fig 3G to the supplementary materials.

Point by point response to reviewer's comments

Reviewer #2 (Remarks to the Author):

1. The revised manuscript improved compared to the previous version. The authors addressed most of the issues raised by the reviewers. However, the new Fig. 3a showing the gel of the pull down experiment is not very clear. To this reviewer the circled regions have been cut out and analyzed by LC/MS, which is kind of logical. But the gel is not labelled appropriately, e.g. the lane next to the ladder also reveals a band migrating similar to the one shown in the next lane (circled band). What has been loaded in this lane.

We agree with the reviewer that the gel electrophoresis run was suboptimal. That reflected the fact that we showed one of the first experiments performed using a quite complex technique. Since then, the techniques have been refined by using chemically synthesized biotinylated aptamers during the pull down and by optimizing masking and washing steps. A new pull-down experiment performed with the optimized technique has been provided in the revised Figure 3A. The new data clearly show a band differentially present in the m12-3773 precipitate compared to the scrambled. The text has been modified to reflect the experiment shown.

2. Likewise, the scrambled control shows a faint band and smear in this area which was also circled cut out. What was identified in this band? How was the gel stained? The methods section does not reveal the staining procedure.

We agree that we were not sufficiently clear in the result section and in the methods. We modified the text in the results by adding on page 15 of the current version the following paragraph "*Briefly, we incubated single cell suspension from human islets with islet specific or scrambled biotinylated aptamers, extensively washed the cells, gently solubilized the cell membrane, recovered the aptamers bound to their target with streptavidin magnetic beads, run a SDS page, and evaluated the differentially expressed proteins by Coomassie blue staining.*"

The list of proteins detected with in the m12-3773 and scrambled precipitates was and is provided in supplementary table 5. Methods on page 35 of the current version have been modified to reflect the new experimental setting and better described the experimental procedure.

3. The gel also do not reveal the streptavidin monomer bands, which shall occur at 15kDa. The overall quality of the gel is poor, signal noise ratio is limited.

Most likely with the previous technique the streptavidin was retained with the beads and prevented its entrance in the gel while the aptamers and the bound protein dissociated from the capture biotinylated oligonucleotides. In the new experimental setting, we do observe a 15Kd band.

Reviewer #4 (Remarks to the Author):

The authors have responded satisfactorily to most of the issues brought up by Reviewer However, the issues raised regarding probe beta cell specificity and affinity were not addressed

sufficiently, and at least the potential impact of these drawbacks should be highlighted in the discussion section.

1. Aptamer beta cell specificity:

Both Apt 1-717 and Apt m12-3773 labels both islets but also some exocrine tissue. As the authors points out in the introduction, novel agents with improved beta cell selectivity is required compared to existing imaging agents. However, the results seen here in eg Fig 1F, Fig 2D, Fig 5A, Suppl Fig 3A-B, Suppl Fig 6 clearly demonstrates labelling of exocrine cells.

We agree with the reviewer that formulation is essential to maximize the specific binding as he mentioned below. Paradoxically, we found more challenging the use of the aptamers in vitro than in vivo. For example, we do observe binding on some acinar mouse cell when we stain pancreas in vitro (supplementary figure 9 in the revised manuscript) but not when we stain the cells in living animals (figure 5d). Notwithstanding that some cells of the acinar tissues may express TMED6 and clusterin (i.e. those close to the duct and possible beta cell precursors expressing also insulin), some of the non-islet signal observed in the in vitro staining can derive by a suboptimal masking of the tissues. Two main issues affect the staining using aptamers chemically labeled with Cyanine 3: a) not every aptamer molecule is labeled with the cy3 fluorochrome and 2) the cy3 molecule can integrate in positions that disrupt aptamer tridimensional structure. Indeed, in some experiments in which we added an average of 3-4 cyanine molecules per aptamer molecule (in the experiments shown we conjugate an average of 1.2 cy3 molecule per aptamer), we lost the specific signal and observed a higher background in throughout the pancreas. This non-specific signal was also detectable in supplementary figure 6 (now supplementary figure 7 in the revised manuscript) and was not removed by the incubation with the recombinant TMED6 protein.

Since the last submission (~1 year ago), we improved the staining by employing biotinylated aptamer and developing better masking agents. These improvements reduced aptamer binding to the acinar tissue and improved the signal to the islets (figure 1 for reviewers). Staining using the improved technique are now depicted in the revised figure 2d and (at lower magnification) in supplementary figure 4 of the revised manuscript. Unfortunately, time and economic considerations do not allow us to repeat with the improved staining techniques all the staining experiments showed in the original manuscript. Thus, we moved the former figure 2d in the supplementary material (Supplementary figure 3c) to keep consistencies with the tissue arrays screening, and scanned images.

Additionally, consistently only a subset of insulin positive cells are stained also with the aptamers. The limitation of the specificity should be discussed more clearly in the manuscript eg how will targeting of some exocrine tissue (and only targeting some beta cells) in affect potential applications.

Any signal from the exocrine pancreas, will most likely mask the in vivo islet specific signal for an imaging probe (as the islets are diluted 50-100 times).

We agree, this is one of the reasons while we propose the use of multiple aptamers each recognizing a different epitope of beta cells to maximize the signal to background ratio. We realized that this concept was not sufficiently discussed. In page 15 of the revised version, we modified the discussion to introduce the concept of combinatorial specificity as follow:

“It is important to note that both aptamer 1-717 and m12-3773 recognize different β cells from at different intensity suggesting a differential expression of their targets within β cell subsets. This suggests the combinatorial use of multiple aptamers each specific for a different β cell specific

epitope to maximize signal to background ratio. The combinatorial use of multiple β cell-specific aptamers should maximize the number of targeted insulin-producing cells, increase the quantity of imaging reagent delivered to the cells expressing both targets, and minimize binding to other tissues and cell expressing only one target.

Both aptamers recognized mouse and human islets *in vitro* and, more importantly, *in vivo* (Figs.4, 5). Our *in vivo* biodistribution experiments (Fig.4) sustain the concept of combinatorial specificity since the signal observed with an equimolar mixture of the two aptamers significantly increased the specific signal in the islet graft and lowered the non-specific signal in other tissues. Indeed, binding of β cells targeting agents to non-insulin producing cells is a problem reported also with other probes. For example, antibodies against GLP1R do recognize pancreatic duct and other cells in the exocrine pancreas and in the thyroid⁶³. Similarly, VMAT2 expression is not completely restricted to β -cells but is expressed also in a fraction of pancreatic polypeptide secreting cells and on pancreatic nerves⁶⁴. The combinatorial use of multiple imaging probes each specific for a marker preferentially (although not exclusively) expressed by β cells has the potential to increase our capacity to accurately measure β cells. Indeed, the systemic administration of an equimolar mixture aptamers 1-717 and m12-3773 yielded a fluorescence signal in the graft region that was proportional to the number of transplanted islets (Fig.4) indicating that the combinatorial use of our aptamers can be used to measure β cell mass. Similar experiments performed in BALB/c mice showed that a mixture of both aptamers can efficiently label the endogenous β cells *in vivo* and monitor the rejection of islet allografts (Fig.5)."

2. Aptamer affinity:

Apt 1-717, which putative target is TMED6, has two different affinities given - in the 0.1 μ M range (Suppl Fig 5, by fluorescent staining) and in the nano molar range (Fig 3G, by SPR).

Here, the SPR fit seems poor (Fig 3G), especially for the dissociation rate (kd) which drives the nanomolar value of the dissociation constant KD. In fact the lowest concentrations are way above the estimated KD. So I would assume the affinity from Suppl Fig 5 is the most reliable, again in the 0.1 μ M range.

A KD in the 0.1 μ M range is definitely not sufficient for *in vivo* PET imaging, especially when attempting to image a diluted tissue like the pancreatic beta cells.

The same is true for Apt m12-3773.

We agree that the differences in the affinity value observed by flow cytometry and SPR deserve few considerations. While it is true that the SPR fit is suboptimal and driven mostly by the dissociation rate, flow cytometry determination of the affinity is affected by the cell death inevitable when using primary cells from dissociated human islets and cy3 labeled aptamers as discussed above (see the review by Hunter and Cochran <https://pubmed.ncbi.nlm.nih.gov/27586327/> that address some of these issues). Additionally, since the aptamers and the fluorochrome penetrate the β cells cytoplasm upon binding *in vivo* (making the dissociation constant from the cells extremely low) the *in vitro* KD value are not necessary indicative of the efficacy of the aptamers as PET probes.

To address the above mentioned technical issues, we determined the KD in complete media using as target the immortalized insulinoma cell line MIN6 and as probe aptamer XIAP chimera in which the guide strand was labeled with cy5 in 5'. This setting drastically reduced the cell death and avoid the above-mentioned limitation associated with cy3 chemically labeled aptamers. The KD measured with this assay were 67 x10⁻⁹ and 46x10⁻⁹ for aptamer m12-3773 and 1-717. These results are now depicted in figure 3e and 3i. Additionally, we performed

similar experiments using biotinylated aptamer complexed with streptavidin. This tetrameric aptamer formulation shows apparent KD of 14 nM and 20 nM for aptamer 1-717 and m12-3773. These results have been included in supplementary figure 6.

Thus, the fact that the relatively poor affinity of the probes (with respect to imaging) will make PET imaging more or less impossible should be addressed in the discussion. E.g. probe improvement is required before further progress in SPECT/PET in vivo imaging. I suggest to show Suppl Fig 5 in the main manuscript and instead move the SPR Fig 3G to the supplementary materials.

Following your suggestion and in line with the above-mentioned new experiments, we included in the discussion (page 16 of the current version) the following paragraph mentioning these results and suggesting the use of multimeric aptamers for the development of SPECT/PET probes.

“SPR and flow cytometry assays showed different affinity of our aptamers for the cognate targets. While SPR and flow cytometry against MIN6 cells indicate a KD in the nanomolar range, flow cytometry against primary human β cells gave an apparent KD of $\sim 1 \times 10^{-7}$ that might not suffice for PET and SPECT probes. Although the inevitable presence of dead cells after islet dissociation can explain this suboptimal affinity for primary human β cells⁶⁵, the future imaging probe might require improvements using for example multimeric forms of our aptamers. “

REVIEWERS' COMMENTS

Reviewer #2 (Remarks to the Author):

The revised version of the manuscript by Van Simaeys et al. addressed all issues raised by the referees and gave reasonable explanations for those issues that could not be addressed experimentally. I therefore recommend acceptance.

Reviewer #4 (Remarks to the Author):

The authors has provided new experimental data that answered my remaining concerns regarding specificity and sensitivity. Especially the new affinity data of the aptamers in relevant in vitro models make the manuscript substantially improved.

Rebuttal to reviewers' comment

REVIEWERS' COMMENTS

Reviewer #2 (Remarks to the Author):

- The revised version of the manuscript by Van Simaeys et al. addressed all issues raised by the referees and gave reasonable explanations for those issues that could not be addressed experimentally. I therefore recommend acceptance.

Thank you

Reviewer #4 (Remarks to the Author):

- The authors has provided new experimental data that answered my remaining concerns regarding specificity and sensitivity. Especially the new affinity data of the aptamers in relevant in vitro models make the manuscript substantially improved.

Thank you